# Scalable Generative Modeling of Weighted Graphs

**Richard Williams**  *rlwilliams34@ucla.edu*
*Department of Biostatistics*
*University of California, Los Angeles*

**Eric Nalisnick**  *nalisnick@jhu.edu*
*Department of Computer Science*
*Johns Hopkins University*

**Andrew Holbrook**  *aholbroo@ucla.edu*
*Department of Biostatistics*
*University of California, Los Angeles*

**Reviewed on OpenReview:** *https://openreview.net/forum?id=yWKkBOcD18&noteId=AQrmkZ9eWM*

## Abstract

Weighted graphs are ubiquitous throughout biology, chemistry, and the social sciences, motivating the development of generative models for abstract weighted graph data using deep neural networks. However, most current deep generative models are designed for unweighted graphs and cannot be easily extended to weighted topologies. Among those that do incorporate edge weights, few consider a joint distribution with the topology of the graph. Furthermore, learning a distribution over weighted graphs must account for complex nonlocal dependencies between both the edges of the graph and corresponding weights of each edge. We develop an autoregressive model BiGG-E, a nontrivial extension of the BiGG model, that learns a joint distribution over weighted graphs while exploiting sparsity to generate a weighted graph with $n$ nodes and $m$ edges in $O((n+m)\log n)$ time. Simulation studies and experiments on a variety of benchmark datasets demonstrate that BiGG-E best captures distributions over weighted graphs while remaining scalable and computationally efficient.

## 1 Introduction

Graphs are useful mathematical structures for representing data in many domains, with applications ranging from modeling protein–protein interactions (Keretsu & Sarmah, 2016) to predicting time spent in traffic (Stanojevic et al., 2018). A graph consists of a set of objects, called nodes, and their corresponding connections, called edges, which represent the graph's topology. Edges may contain additional information in the form of edge features, which can be categorical – such as bond types in molecular graphs (Jo et al., 2022) – or continuous – such as branch lengths in phylogenetic trees (Semple et al., 2003). Edge weights, in particular, are continuous single-dimensional edge features, and a graph with edge weights comprises a weighted graph. Weighted graphs hold many applications, such as in neuroscience (Barjuan et al., 2025), economics (Fagiolo et al., 2010), social networks (Bellingeri et al., 2023), and phylogenetics (Baele et al., 2025).

Generative modeling of graphs is a vibrant research area, where weighted graphs require models that capture both topology and edge-weight distributions. Early approaches such as Erdős–Rényi (Erdős & Rényi, 1959) and Barabási–Albert models (Albert & Barabasi, 2002) offer simple graph-generating mechanisms but fail to capture subtle dependencies observed between edges in real-world data. These limitations motivate the development of expressive deep generative models capable of learning complex, nonlinear relationships. Despite such advances, modeling graph distributions remains an ongoing challenge due to the combinatorial nature of graphs and complex dependencies among edges. Furthermore, although incorporating edge weights seems straightforward, jointly modeling discrete topology and continuous weights introduces additional complexity, requiring the model to account for dependencies both within and between these two components.

Modern graph generative models – including variational autoencoders (VAEs) (Kipf & Welling, 2016; Grover et al., 2019), graph neural networks (Grover et al., 2019), autoregressive models (You et al., 2018; Liao et al., 2019; Li et al., 2018; Dai et al., 2020), and score-based diffusion models (Niu et al., 2020; Jo et al., 2022; Vignac et al., 2023) – primarily focus on unweighted graphs. Most limit their scope to modeling distributions over graph topology while offering limited insight into the joint modeling of topology and edge weights, and few provide scalable solutions for learning joint distributions over sparse weighted graphs. Furthermore, a significant computational bottleneck in graph generative modeling arises when jointly modeling all possible edge connections, which scales quadratically with the number of nodes. Hence, models that attempt to jointly model all possible edge connections are computationally slow and infeasible for even moderately sized graphs. Autoregressive models factorize graph generation node-by-node using a sequential decision process. BiGG, "Big Graph Generation" (Dai et al., 2020), augments this approach by directly generating the edge set of sparse graphs, scaling to graphs with tens of thousands of nodes. However, existing autoregressive methods, including BiGG, remain limited to unweighted graphs.

To address the need for efficient generative modeling over large weighted graphs, we introduce BiGG-E ("BiGG-Extension"), an autoregressive model that jointly generates both graph topologies and edge weights while preserving the scalability of its unweighted predecessor, BiGG. We benchmark BiGG-E against three alternatives: (1) Adjacency-LSTM (Adj-LSTM), a fully expressive but computationally inefficient model parameterized with a Long Short-Term Memory (LSTM; Hochreiter & Schmidhuber (1997)) cell; (2) BiGG-MLP, a naive extension that appends encodings of weights to BiGG using a multilayer perceptron (MLP, Rumelhart et al. (1986)); and (3) BiGG+GCN, a two-stage model that decouples topology and weight generation.

Our contributions are as follows:

- We propose BiGG-E, an application-agnostic generative model that learns joint distributions over sparse weighted graphs.
- We empirically demonstrate that BiGG-E maintains the efficient scaling of BiGG while outperforming BiGG-MLP, Adj-LSTM, and BiGG+GCN.
- All BiGG extensions are orders of magnitude faster than Adj-LSTM and SparseDiff, a diffusion-model competitor.
- We directly evaluate the joint and marginal generative performance of all models on an array of weighted graph distributions.

## 2 Background

### 2.1 Data

Let $\mathcal{G} = \{G_1, \ldots, G_{|\mathcal{G}|}\}$ be an independent sample of weighted graphs from an unknown data-generating distribution $p(G_i)$, for $i = 1, \ldots, |\mathcal{G}|$. Each weighted graph is defined as $G_i = (V_i, E_i, W_i)$, where $V_i = \{v_1, \ldots, v_{n_i}\}$ is the set of $|V_i| = n_i$ nodes, $E_i \subseteq V_i \times V_i$ is the set of $|E_i| = m_i$ edges, and $W_i : V_i \times V_i \to \mathbb{R}^+$ maps edges to positive edge weights. For notational simplicity, we drop the subscript $i$, under the assumption that the graphs $G_i \in \mathcal{G}$ are independent and identically distributed.

For any edge $(v_i, v_j) \in E$, the edge weight is $W(v_i, v_j) = w_{ij}$; otherwise, it is zero. The weighted adjacency matrix $\mathbf{W} \in \mathbb{R}^{n \times n}$ has entries $W(v_i, v_j)$. In the unweighted case, edge weights are 1 when $(v_i, v_j) \in E$ and 0 otherwise. We denote the unweighted adjacency matrix by $\mathbf{A}$, which encodes the graph's topology, and use $\mathbf{W}$ to specifically refer to the weighted adjacency matrix.

A weighted graph $G$ under node ordering $\pi$ is represented by its permuted weighted adjacency matrix $\mathbf{W}^{\pi}$, from which the probability of observing $G$ is given by $p(G) = p(|V| = n) \sum_{\pi} p(\mathbf{W}^{\pi(G)})$ (Dai et al., 2020). Because summing over all $n!$ node permutations quickly becomes intractable, we follow Liao et al. (2019) and assume a single canonical ordering $\pi$, yielding the lower bound estimate $p(G) \simeq p(|V| = n)p(\mathbf{W}^{\pi(G)})$. Following Dai et al. (2020), we estimate $p(|V| = n)$ using a multinomial distribution over node counts in the training set, and model $p(\mathbf{W}^{\pi(G)})$ with deep autoregressive neural networks parameterized by $\boldsymbol{\theta}$. We assume all graphs are under the canonical ordering $\pi(G)$ and omit this notation moving forward.

## 2.2 Related Work

**Weighted Graph Generative Models**  Although various models incorporate edge and node features in the graph generative process, these features are typically categorical (Kipf & Welling, 2016; Li et al., 2018; Kawai et al., 2019) or are tailored to a specific class of graphs, such as protein graphs (Ingraham et al., 2019). Furthermore, previous work on autoregressive models (You et al., 2018; Liao et al., 2019; Dai et al., 2020) focuses exclusively on unweighted graphs. Most implementations on weighted graphs provide limited insight into the incorporation of edge weights. Graphite (Grover et al., 2019) proposes modeling weighted graphs by parameterizing a Gaussian random variable, which introduces the possibility of infeasible negative weights. Although score-based models incorporate edge features into the graph generative process, these features are typically categorical (Vignac et al., 2023) or rely on thresholding to produce a weighted adjacency matrix $\mathbf{W}$ and only evaluate performance on the binarized adjacency matrix (Niu et al., 2020).

**Scalability**  Scaling generative models to graphs with thousands of nodes is an ongoing challenge, as the adjacency matrix $\mathbf{A}$ has $\mathcal{O}(n^2)$ entries. In addition, many VAE (Grover et al., 2019) and diffusion (Niu et al., 2020; Vignac et al., 2023) models utilize graph neural networks, which perform convolutions over the entire adjacency matrix of the graph. SparseDiff (Qin et al., 2024) is a scalable diffusion model on sparse graphs, but only scales to graphs with hundreds of nodes, while we are interested in scaling to thousands of nodes.

Autoregressive models currently scale best with large graphs. While GraphRNN (You et al., 2018) trains in $\mathcal{O}(n^2)$ time despite using a breadth-first search ordering scheme to reduce computational overhead, GRAN (Liao et al., 2019) trains in $\mathcal{O}(n)$ time by generating blocks of nodes of the graph at a time, but trades this gain in scalability for worsened sample quality as the model estimates edge densities per block of nodes. BiGG (Dai et al., 2020) leverages the sparsity of many real-world graphs and directly generates the edge set $\{e_k\}$ of $\mathbf{A}$:

$$p_{\boldsymbol{\theta}}(\mathbf{A}) = \prod_{k=1}^{m} p_{\boldsymbol{\theta}}(e_k | \{e_{l:l<k}\}). \tag{1}$$

BiGG trains on the order $\mathcal{O}(\log n)$ time, generates an unweighted graph in $\mathcal{O}((n + m) \log n)$ time, and scales to graphs with up to 50K nodes. Currently, BiGG and other autoregressive models remain limited to unweighted graphs, precluding the sampling of edge weights. These limitations motivate the need for a scalable autoregressive model capable of modeling joint distributions over weighted graphs.

# 3 Methods and Contributions

## 3.1 Joint Modeling of Topology and Edge Weights

Previous autoregressive models produce unweighted graphs either by directly generating $\mathbf{A}$ (You et al., 2018), or by directly generating the edge set (Dai et al., 2020). However, our models learn over *weighted* adjacency matrices $\mathbf{W}$. As such, we first define a joint distribution over the existence of an edge $e$ and its corresponding edge weight $w$. To do so, note that as $w$ is only sampled when $e$ exists, we can naturally factor the joint probability $p_{\boldsymbol{\theta}}(e, w)$ of observing a weighted edge $(e, w)$ as

$$p_{\boldsymbol{\theta}}(e, w) = p_{\boldsymbol{\theta}}(e) p_{\boldsymbol{\theta}}(w | e), \tag{2}$$

where $p_{\boldsymbol{\theta}}(e)$ is the parameterized Bernoulli probability that an edge exists between two nodes, and $p_{\boldsymbol{\theta}}(w | e)$ is the probability of drawing a corresponding weight given that $e$ exists. Since $w$ is assumed to be continuous, let $p_{\boldsymbol{\theta}}(w | e)$ represent the distribution of the random variable $w$ with parameterized density $f_{\boldsymbol{\theta}}(w)$. In the case where no edge exists and $e = 0$, set $w = 0$ with probability 1; otherwise, if an edge exists and $e = 1$, draw a corresponding weight from a conditional distribution $p_{\boldsymbol{\theta}}(w | e)$.

We parameterize the conditional distribution $p_{\boldsymbol{\theta}}(w | e)$ as a normal random variable $\epsilon | e \sim N(\mu, \sigma)$ transformed with the softplus function $\text{Softplus}(\epsilon) = \log(1 + \exp(\epsilon))$. In our experience, such a transformation of a

normal random variable performs best with gradient-based optimization by providing enough flexibility in modeling distributions, where work such as Rodríguez & Dunson (2011) empirically demonstrates that a probit transformation of a random normal variable provides a prior capable of generating a rich class of distributions. To ensure positivity of the weights, the softplus function maps each value from the normal distribution to a positive real number. Other candidate distributions, such as the gamma and log-normal distributions, are more challenging to implement because of the complexity of the likelihood in the former and the heavy right-tailedness in the latter. Thus, with the softplus-normal conditional density placed on the weights, the term $p_{\boldsymbol{\theta}}(w|e)$ in Equation 2 is equal to

$$p_{\boldsymbol{\theta}}(w|e) \propto \frac{1}{2\sigma^2} \exp\left[ -\frac{1}{2\sigma^2} \big(\log(e^w - 1) - \mu\big)^2 \right] \tag{3}$$

up to a constant factor, where $\mu$ and $\sigma^2$ are functions of neural network parameters $\boldsymbol{\theta}$.

## 3.2 Likelihood of a Weighted Adjacency Matrix

There are two ways to parameterize the distribution $p_{\boldsymbol{\theta}}(\mathbf{W})$ over weighted graphs: first, we may consider the probability over all entries of $\mathbf{W}$ in a row-wise manner as

$$p_{\boldsymbol{\theta}}(\mathbf{W}) = \prod_{i=1}^{n}\prod_{j=1}^{i-1} p_{\boldsymbol{\theta}}(W_{ij}|\{W_{kl}\}) = \prod_{i=1}^{n}\prod_{j=1}^{i-1} (1 - p_{ij})^{1-e_{ij}} \big[p_{ij} p_{\boldsymbol{\theta}}(w_{ij}|e_{ij})\big]^{e_{ij}}, \tag{4}$$

where $W_{ij}$ is the $(i, j)$-th entry of $\mathbf{W}$, $p_{ij} \equiv p_{ij}(\boldsymbol{\theta})$ is the estimated probability of an edge existing between nodes $v_i$ and $v_j$, $e_{ij} = 1$ when $(v_i, v_j) \in E$ and is otherwise 0, and $w_{ij} = W(v_i, v_j)$ is the weight of edge $e_{ij}$ whenever $(v_i, v_j) \in E$. Note each entry $W_{ij}$ is conditioned on all prior entries, denoted as $\{W_{kl}\}$.

Next, similarly to how BiGG factors $p_{\boldsymbol{\theta}}(\mathbf{A})$ in Equation 1, we factor the weighted edge set of $\mathbf{W}$ as

$$p_{\boldsymbol{\theta}}(\mathbf{W}) = \prod_{k=1}^{m} p_{\boldsymbol{\theta}}(e_k|\{(e_l, w_l)_{:l<k}\}) \cdot p_{\boldsymbol{\theta}}(w_k|e_k, \{(e_l, w_l)_{:l<k}\}), \tag{5}$$

noting all edges up to and including $e_k$ condition the generation of weight $w_k$. We substitute Equation 3 into the terms $p_{\boldsymbol{\theta}}(w_{ij}|e_{ij})$ and $p_{\boldsymbol{\theta}}(w_k|e_k, \{(e_l, w_l)_{:l<k}\})$ of Equations 4 and 5, respectively, and maximize the log-likelihood $\mathcal{L}(\boldsymbol{\theta}; \mathbf{W})$ to train our models on weighted graphs. More details of the derivation of Equations 4 and 5 and the objective function are given in Appendix A.1.

## 3.3 Models

Our main contributions use autoregressive models, which are well-suited for graph generation as they explicitly capture dependencies among edges and, in our case, their weights. BiGG-E extends the original BiGG model by maintaining two states during generation: a topological state inherited from BiGG and a weight state that encodes all previously generated edge weights. By leveraging both states, BiGG-E jointly and autoregressively predicts the weighted edge set of $\mathbf{W}$. We begin by reviewing the BiGG architecture before detailing our extensions in BiGG-E. Additional architectural details and an expanded BiGG review are provided in Appendices A.2 and A.2.1.

### 3.3.1 Review of BiGG (Dai et al., 2020)

BiGG generates an unweighted graph with an algorithm consisting of two main components, both of which train in $\mathcal{O}(\log n)$ time: (1) row generation, where BiGG generates each row of the lower half of $\mathbf{A}$ using a binary decision tree; and (2) row conditioning, where BiGG deploys a hierarchical data maintenance structure called a Fenwick tree (Fenwick, 1994) to condition the subsequent row generation on all previous rows.

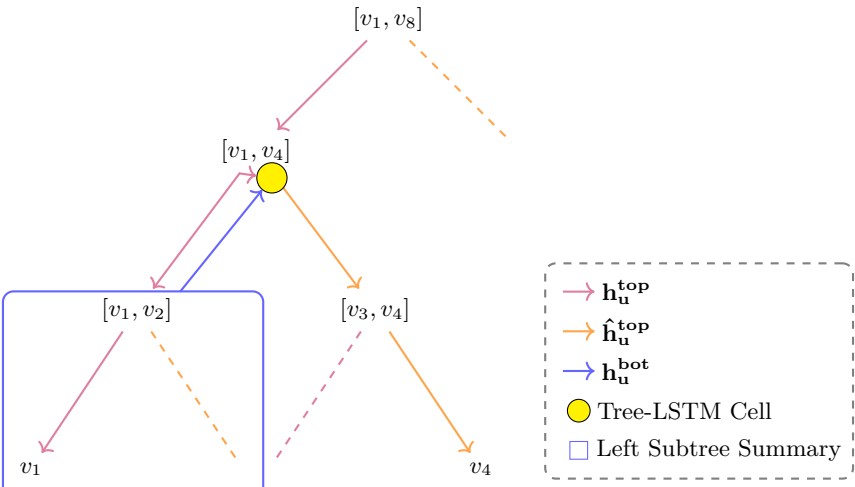

Figure 1: Example of constructing $\mathcal{T}_u$ in the original BiGG model. The interval $[v_1, v_8]$ is recursively partitioned via left-right decisions until reaching individual nodes (e.g., $v_1$ and $v_4$). Dashed lines indicate no edge. Purple arrows show the top-down context vector $\mathbf{h}_u^{\text{top}}$ used for left-child edge predictions; orange arrows show the conditioned top-down context $\hat{\mathbf{h}}_u^{\text{top}}$ used for right-child predictions. At node $[v_1, v_4]$, a TreeLSTM merges the left sub-tree summary (blue) with $\mathbf{h}_u^{\text{top}}$ to produce $\hat{\mathbf{h}}_u^{\text{top}}$ for left subtree conditioning.

**Row Generation** To sample edge connections for each node $v_u \in G$, BiGG adapts a procedure from R-MAT (Chakrabarti et al., 2004) to construct a binary decision tree $\mathcal{T}_u$, which identifies all edge connections with $v_u$ by recursively partitioning the candidate edge interval $[v_1, v_{u-1}]$ into halves. Each node $t \in \mathcal{T}_u$ corresponds to a subinterval $[v_i, v_k]$ of length $l_t = k - i + 1$, which is split into left and right halves: $\text{lch}(t) = [v_i, \ v_{i+\lfloor l_t/2 \rfloor}]$ and $\text{rch}(t) = [v_{i+\lfloor l_t/2 \rfloor+1}, \ v_k]$.

If an edge exists in $\text{lch}(t)$, the model recurses into that interval until reaching a singleton interval $[v_j, v_j]$, which represents an edge connection with $v_j$. After completing the left subtree, the model recurses into $\text{rch}(t)$, conditioned on all edge connections – if any – formed in $\text{lch}(t)$. BiGG conditions subsequent predictions on all prior interval splits using two context vectors: (1) predictions for $\text{lch}(t)$ use a top-down hidden state $\mathbf{h}_u^{\text{top}}(t)$, which sequentially encodes all left and right edge existence decisions made in $\mathcal{T}_u$ thus far; and (2) predictions for $\text{rch}(t)$ use a conditioned top-down hidden state $\hat{\mathbf{h}}_u^{top}(t)$ computed by merging $\mathbf{h}_u^{\text{top}}(t)$ with a bottom-up summary state of the generated left subtree, $\mathbf{h}_u^{\text{bot}}(\text{lch}(t))$. This merge is performed using a Tree-LSTM Cell (Tai et al., 2015), which encodes relevant information from the top-down and bottom-up hidden states: $\hat{\mathbf{h}}_u^{top}(t) = \text{TreeCell}_{\boldsymbol{\theta}}(\mathbf{h}_u^{\text{top}}(t), \mathbf{h}_u^{\text{bot}}(\text{lch}(t)))$.

Hence, constructing $\mathcal{T}_u$ is fully autoregressive with probability

$$p_{\boldsymbol{\theta}}(\mathcal{T}_u) = \prod_{t \in \mathcal{T}_u} p_{\boldsymbol{\theta}}(\text{lch(t)}|\mathbf{h}_u^{\text{top}}(t)) \cdot p_{\boldsymbol{\theta}}(\text{rch(t)}|\hat{\mathbf{h}}_u^{\text{top}}(t)). \tag{6}$$

Figure 1 illustrates an example of constructing $\mathcal{T}_u$ and visualizes use of the top-down and bottom-up states in predicting $\text{lch}(t)$ and $\text{rch}(t)$. Finally, note that as $\mathbf{h}_u^{\text{bot}}(t)$ is the bottom-up summary state summarizing the subtree rooted at node $t$, the bottom-up summary state at the root node $t_0$ summarizes the entire tree $\mathcal{T}_u$.

**Fenwick Tree** To condition each decision tree $\mathcal{T}_u$ on all prior trees $\mathcal{T}_1$ to $\mathcal{T}_{u-1}$, BiGG adopts the Fenwick tree (Fenwick, 1994) to efficiently summarize all previously generated rows in $\mathbf{A}$. The Fenwick tree at row $u$ has $\lfloor \log(u-1) \rfloor + 1$ levels, where the base level (leaves) of the tree are states $\mathbf{h}_u^{bot}(t_0)$ summarizing the edge connections formed in each $\mathcal{T}_j, j = 1, \ldots, u - 1$. Higher levels of the Fenwick tree merge these states

---

**Algorithm 1** BiGG-E Weight Sampling and Embedding

---

**function** embed_weight($w_k, \mathbf{h}^{\text{wt}}_{k-1}$)

1: $\mathbf{w}^0_k = \text{LSTM}_{\boldsymbol{\theta}}(w_k)$

2: Add $\mathbf{w}^0_k$ to Fenwick weight tree and update tree using Equation 7.

3: $\mathbf{h}^{\text{wt}}_k = \text{TreeCell}^{\text{summary}}_{\boldsymbol{\theta}}\left(\left[\mathbf{w}^i_{\lfloor \frac{k}{2^i} \rfloor} \text{ where } k \ \& \ 2^i = 2^i\right]\right)$

4: Return $\mathbf{h}^{\text{wt}}_k$

**end function**

**function** sample_weight($u, t, \mathbf{h}^{\text{top}}_u(t), \mathbf{h}^{\text{wt}}_k$)

5: $\mathbf{h}^{\text{sum}}_{u,k}(t) = \text{TreeCell}^{\text{merge}}_{\boldsymbol{\theta}}(\mathbf{h}^{\text{top}}_u(t), \mathbf{h}^{\text{wt}}_k)$

6: Set $\mu_{k+1} = f_\mu(\mathbf{h}^{\text{sum}}_{u,k}(t))$ and $\log \sigma^2_{k+1} = f_{\sigma^2}(\mathbf{h}^{\text{sum}}_{u,k}(t))$

7: Sample $w_{k+1}$ from Section 3.1 using $\mu_{k+1}, \sigma^2_{k+1}$.

8: $\mathbf{h}^{\text{wt}}_{k+1} = \text{embed\_weight}(w_{k+1}, \mathbf{h}^{\text{wt}}_k)$

9: Return $\vec{1}$, {edge index $t$ represents}, $w_{k+1}, \mathbf{h}^{\text{wt}}_{k+1}$

**end function**

---

to produce aggregated summaries across multiple rows. Letting $\mathbf{g}^i_j$ denote the state at the $j$-th node on the $i$-th level, each non-leaf node of the Fenwick tree merges its two children using a Tree-LSTM cell as

$$\mathbf{g}^i_j = \text{TreeCell}^{\text{row}}_{\boldsymbol{\theta}}(\mathbf{g}^{i-1}_{2j-1}, \mathbf{g}^{i-1}_{2j}), \tag{7}$$

where $1 \leq i \leq \lfloor \log(u-1) \rfloor + 1$, $1 \leq j \leq \lfloor \frac{u}{2^i} \rfloor$, and $\mathbf{g}^0_j$ is the bottom-up summary state of $\mathcal{T}_j$. Finally, to summarize all rows 1 to $u - 1$, the model iteratively applies a Tree-LSTM cell to produce a row summary hidden state $\mathbf{h}^{\text{row}}_u$:

$$\mathbf{h}^{\text{row}}_u = \text{TreeCell}^{\text{summary}}_{\boldsymbol{\theta}}\left(\left[\mathbf{g}^i_{\lfloor \frac{u}{2^i} \rfloor} \text{ where } u \ \& \ 2^i = 2^i\right]\right), \tag{8}$$

where $\&$ is the bit-level 'and' operator, each $\mathbf{g}^i_{\lfloor \frac{u}{2^i} \rfloor}$ encodes summaries of different groups of rows of $\mathbf{A}$, and $\mathbf{h}^{\text{row}}_u$ initializes $h^{\text{top}}_u(t_0)$ to condition construction of $\mathcal{T}_u$ so that $p_{\boldsymbol{\theta}}(\mathcal{T}_u) \equiv p_{\boldsymbol{\theta}}(\mathcal{T}_u|\mathcal{T}_1, \ldots, \mathcal{T}_{u-1})$.

**BiGG Training and Sampling Times** The training procedure for BiGG consists of four steps, each running in $\mathcal{O}(\log n)$ time by parallelizing computations across rows. First, since trees $\mathcal{T}_u$ are summarized independently, their root-level summaries are computed in parallel by traversing each tree level by level from the leaves to the root. Second, the Fenwick tree is constructed from these root summaries in the same level-wise manner. Third, the model computes all row summaries $\mathbf{h}^{\text{row}}_u$ using the Fenwick tree. Finally, for each $\mathcal{T}_u$, the model computes all left and right edge interval existence probabilities level-by-level.

For graph generation, the Fenwick tree requires $\mathcal{O}(n \log n)$ time to construct, since updates now occur sequentially across row. Provided the graph is sparse, i.e., $m = \mathcal{O}(n)$, the construction of all trees $\mathcal{T}_u$ requires $\mathcal{O}(m \log n)$ time. Thus, the total sampling time of a sparse unweighted graph is $\mathcal{O}((n + m) \log n)$.

### 3.4 BiGG-E

BiGG-E incorporates a weight state used in tandem with the original topology state of the BiGG model to jointly predict weighted edges. We first describe the weight state, followed by the joint prediction framework.

#### 3.4.1 Edge Weight Prediction

**Constructing the Weight State** To preserve BiGG's training and sampling speed-ups, BiGG-E introduces a second Fenwick data structure – referred to as the Fenwick *weight* tree – that summarizes edge

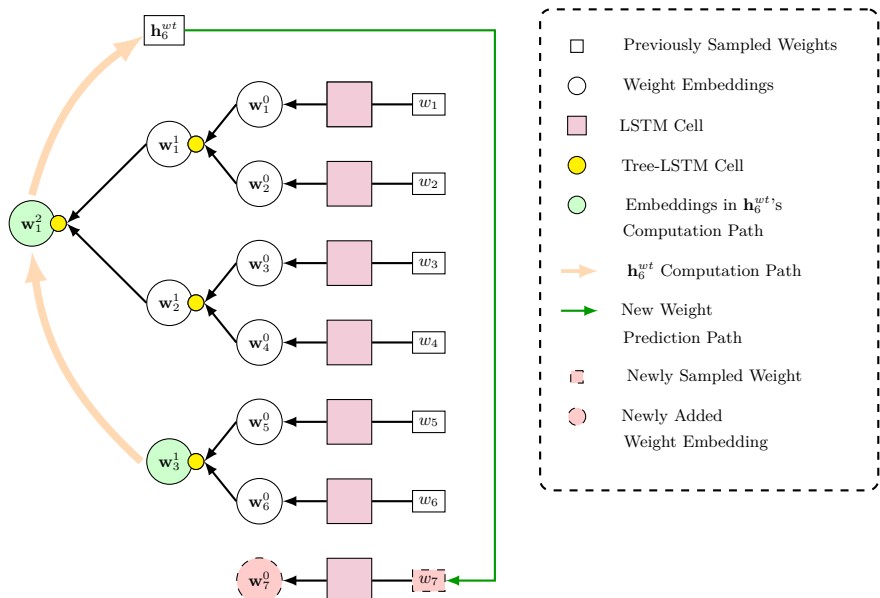

Figure 2: Illustration of Fenwick weight tree state construction. Sampled weights $w_j$ are passed through an LSTM to obtain initial embeddings $\mathbf{w}_j^0$, which are recursively merged using Tree-LSTM cells (yellow nodes) to form higher-level summaries. To compute the current weight hidden state $\mathbf{h}_6^{wt}$, summaries for $w_1$ to $w_4$ ($\mathbf{w}_1^2$) and $w_5$ to $w_6$ ($\mathbf{w}_1^2$) are merged. This state is used to predict $w_7$, which is added to the Fenwick weight tree and updated accordingly.

weights during generation. This new tree mirrors the structure of the original Fenwick *topology* tree used in BiGG, but is maintained separately to construct edge weight embeddings autoregressively. The Fenwick weight tree is similarly organized into $\lfloor \log(k-1) \rfloor + 1$ levels, where $k$ is the current number of weights in the graph and the 0-th level corresponds to initialized weight embeddings, computed using a single forward pass of an LSTM: $\mathbf{w}_k^0 = \text{LSTM}_{\boldsymbol{\theta}}(w_k)$. Higher-level embeddings $\mathbf{w}_j^i$ in the Fenwick weight tree are computed using Equation 7, where $\mathbf{w}_j^0$ is now the initial embedding of the $j$-th weight in the graph.

To obtain a summary state of all prior edge weights, we use Equation 8 to compute the summary weight state $\mathbf{h}_k^{wt}$ for weights $w_1$ to $w_k$ in $\mathcal{O}(\log k)$ steps. Algorithm 1 outlines the edge weight embedding procedure in the Fenwick weight tree. Figure 2 illustrates an example of the Fenwick weight tree embedding process, which updates the weight state recursively based on the weights generated so far.

**Edge Weight Conditioning**   Because the summary weight state $\mathbf{h}_k^{wt}$ encodes information about all previously generated weights $w_1$ to $w_k$, using this state to predict the next edge weight $w_{k+1}$ allows BiGG-E to condition each new weight on the history of prior weights. As established in Section 3.1, each edge weight is sampled from a softplus normal distribution with mean $\mu_k$ and variance $\sigma_k^2$ parameterized by functions of $\boldsymbol{\theta}$. Computing these parameters from $\mathbf{h}_k^{wt}$ in the sampling of the next weight $w_{k+1}$ allows for conditioning $\mu_{k+1}$ and $\sigma_{k+1}^2$ on all preceding weights:

$$\mu_{k+1} = f_\mu(\mathbf{h}_k^{wt}) \qquad \log \sigma_{k+1}^2 = f_{\sigma^2}(\mathbf{h}_k^{wt}) \qquad (9)$$

where $f_\mu$ and $f_{\sigma^2}$ are MLPs that output the estimated mean and log-scale variance, respectively. While this allows BiGG-E to model dependencies among weights, a full generative model must also capture how edge structure and weights influence one another – a task requiring both the topology and weight states.

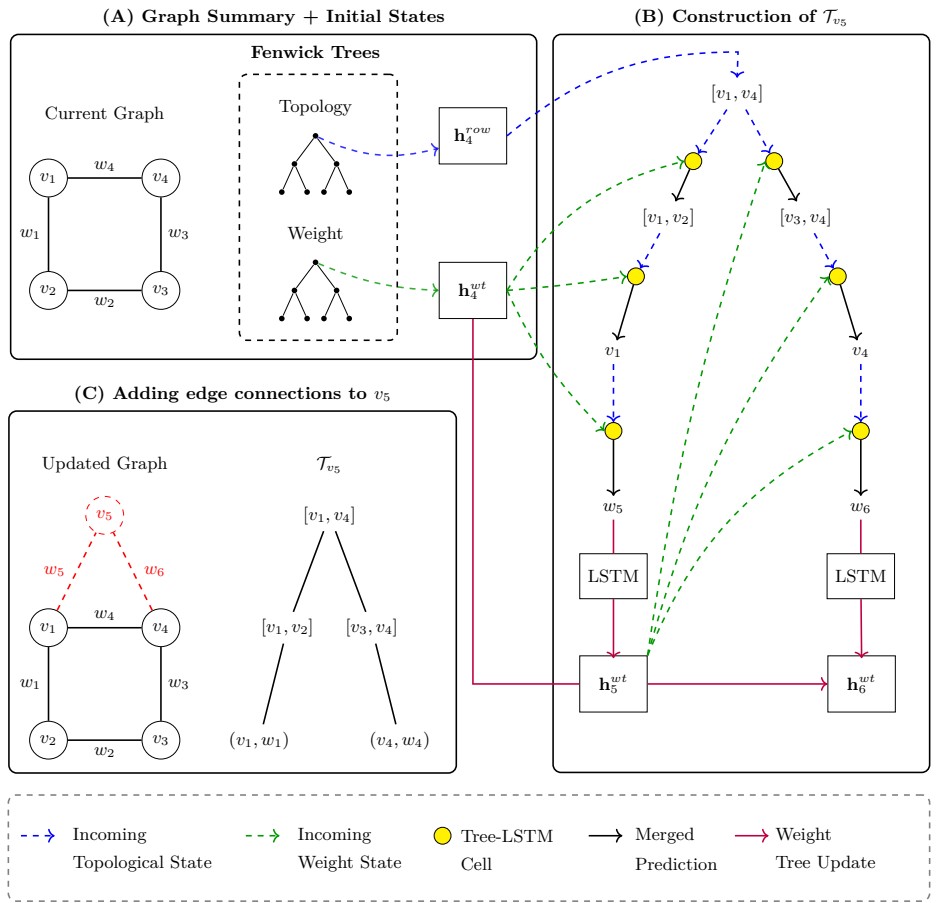

Figure 3: Illustration of the autoregressive construction of $\mathcal{T}_{v_5}$ with weighted edges. Blue dashed arrows indicate the topological state; green dashed arrows indicate the weight state. These are merged with Tree-LSTM cells (yellow) to output hidden states for predicting new weighted edges. New weights are embedded using an LSTM and then added to the Fenwick weight tree. Panels (A–C) depict the process from graph summarization to final integration of $v_5$ into the graph.

### 3.4.2 Joint Modeling

Currently, the topology state is responsible for conditioning edge formation, while the weight state conditions edge weights. Used independently, each state informs only its respective component. However, by leveraging both states during prediction, BiGG-E models the joint interaction between topology and weights. Let $\mathbf{h}_u^{\text{top}}(t)$ be the current top-down context at node $t \in \mathcal{T}_u$ and $\mathbf{h}_k^{\text{wt}}$ be the current weight state of the most recent weight $w_k$. BiGG-E computes the weighted graph summary state $\mathbf{h}_{u,k}^{\text{sum}}(t)$ using a Tree-LSTM cell:

$$\mathbf{h}_{u,k}^{\text{sum}}(t) = \text{TreeCell}_{\boldsymbol{\theta}}^{\text{merge}}(\mathbf{h}_u^{\text{top}}(t), \mathbf{h}_k^{\text{wt}}). \tag{10}$$

First, when constructing $\mathcal{T}_u$, BiGG-E merges $\mathbf{h}_k^{\text{wt}}$ with the top-down context vectors $\mathbf{h}_u^{\text{top}}(t)$ and $\hat{\mathbf{h}}_u^{\text{top}}(t)$ before making predictions for $\text{lch}(t)$ and $\text{rch}(t)$, respectively, using Equation 10. This modifies Equation 6 to

$$p_{\boldsymbol{\theta}}(\mathcal{T}_u) = \prod_{t \in \mathcal{T}_u} p_{\boldsymbol{\theta}}(\text{lch}(t)|\mathbf{h}_{u,k}^{\text{sum}}(t)) \cdot p_{\boldsymbol{\theta}}(\text{rch}(t)|\hat{\mathbf{h}}_{u,k}^{\text{sum}}(t)),$$

which enables further conditioning the graph's topology on the corresponding edge weights.

Next, when BiGG-E forms an edge connection at a singleton interval $t = [v_j, v_j]$ in $\mathcal{T}_u$, the model merges $\mathbf{h}_u^{\mathrm{top}}(t)$ with $\mathbf{h}_k^{\mathrm{wt}}$ using Equation 10 prior to outputting the mean and variance parameters $\mu_{k+1}$ and $\sigma_{k+1}^2$. This modifies Equation 9 to

$$\mu_{k+1} = f_\mu(\mathbf{h}_{u,k}^{\mathrm{sum}}(t)) \qquad \log \sigma_{k+1}^2 = f_{\sigma^2}(\mathbf{h}_{u,k}^{\mathrm{sum}}(t))$$

which conditions sampling edge weights on the topology of the graph, fully allowing BiGG-E to jointly model weighted graphs. Algorithm 1 outlines sampling a weight when an edge exists, and Figure 3 visualizes the joint modeling of weighted edges when constructing $\mathcal{T}_u$.

### 3.4.3 BiGG-E Training and Sampling Time

The novel architectural design outlined in Section 3.4 allows BiGG-E to extend the efficient training and sampling observed from BiGG to sparse weighted graphs. Constructing the weight state with the Fenwick weight tree adds only $\mathcal{O}(\log m)$ training time and $\mathcal{O}(m \log m)$ sampling time, which preserves the overall asymptotic complexity of the model so long as $m = \mathcal{O}(n)$. Furthermore, since BiGG-E constructs each tree $\mathcal{T}_u$ using BiGG's original procedure, training remains parallelizable across rows. Finally, the use of Tree-LSTM cells to merge topological and weight states does not disrupt this structure, allowing BiGG-E to maintain a training time of $\mathcal{O}(\log n)$ and sampling time of $\mathcal{O}((n + m) \log n)$ for sparse weighted graphs.

## 3.5 Comparison Models

Our code for BiGG-E and all comparison models is available at https://github.com/rlwilliams34/BiGG-E.

**Adj-LSTM** Adj-LSTM builds on the work of Li et al. (2018) by using an LSTM cell to parameterize the lower half of $\mathbf{W}$ in a row-wise fashion. To adapt to the grid-like structure of $\mathbf{W}$, Adj-LSTM maintains and updates row and column states autoregressively. When an edge exists between nodes $v_i$ and $v_j$, a corresponding weight is sampled as described in Section 3.1. Adj-LSTM is fully expressive but slow: generating all of $\mathbf{W}$ requires $\mathcal{O}(n^2)$ computations. Additional details are provided in Appendix A.3.

**BiGG-MLP** BiGG-MLP is a simple extension of BiGG that replaces the topology leaf state (Algorithm 1, line 9) that indicates edge existences in $\mathcal{T}_u$ with an MLP that encodes the newly sampled weight into a state embedding. The rest of the algorithm remains unchanged, maintaining a single state that summarizes topology and edge weights. However, using the same state to simultaneously make left-right binary decisions and generate continuous weights severely limits BiGG-MLP's capacity to learn each task. This comparison highlights the importance of maintaining separate topology and weight states, as done in BiGG-E.

**BiGG+GCN** BiGG+GCN is a two-stage model that decouples topology generation from edge weight sampling. First, the original BiGG model generates an unweighted graph. Then, a graph convolutional network (GCN) conditioned on the graph's topology populates each edge with edge weights. Thus, BiGG+GCN forgoes jointly modeling weighted edges and instead generates edges independently of weights. BiGG+GCN serves as a comparison for evaluating the benefits of BiGG-E's joint modeling approach.

**Baseline** An Erdős–Rényi (ER) model independently samples edges with an estimated global edge existence probability. Weights are sampled independently with replacement from the training set edge weights.

## 4 Experiment

We assess all models on the following: (1) do the models capture diverse distributions over weighted graphs; (2) does jointly modeling topology and edge weights improve performance; and (3) do the models scale well to large graphs? Following the evaluation protocol of Liao et al. (2019), we compare an equal number of graphs generated by each model against a held-out test set.

To assess generative quality, we use metrics that evaluate the marginal distributions of graph topology and edge weights, and metrics that evaluate their joint distribution. For topology, we compute Maximum Mean

Discrepancy (MMD) (Gretton et al., 2012) using test functions based on degree distributions, clustering coefficients distributions, and the spectrum of the normalized unweighted Laplacian, which are conventional measures of topological performance used in Liao et al. (2019), Dai et al. (2020), and You et al. (2018). The lower the MMD metric, the higher in quality that graph set is relative to the test set. For all tree and lobster datasets, we also report the error rate as the percent of generated graphs that do not follow the correct structure. For edge weights, we summarize the marginal distribution using the pooled mean and standard deviation, denoted $\bar{w}$ and $s_w$. We additionally compute the MMD on the marginal weight distribution within each graph. To assess joint structure, we apply MMD with test functions based on the weighted Laplacian spectrum and weighted degree distribution. Finally, we compute the average rank of each model within and across all datasets. Further details on the MMD test functions are provided in Appendix A.6.

To evaluate scalability on larger graphs, we train on sets of 80 weighted HW-trees with sizes from {100, 200, 0.5K, 1K, 2K, 5K, 10K, 15K}. Each model is trained on a GV100 GPU with 3.2 GB of memory an single precision performance, and we report the MMD on the normalized Laplacian of the resulting weighted graphs against 20 test graphs. For each trained model, we record the time to sample one graph, the time to complete a forward pass, backward pass, and optimizer step on one training graph, and the memory consumption used per graph during training. We also compare scalability with that of the diffusion model SparseDiff (Qin et al., 2024), hypothesizing that BiGG-E scales more efficiently. Finally, we expect the extended BiGG models to scale better than both Adj-LSTM and SparseDiff, while BiGG-E will retain superior generative quality on large graphs.

**Data**   We use the following datasets to evaluate the generative quality of our models.

- **Erdős–Rényi**: 100 graphs that represent a null case to test whether the models capture the distribution of weighted graphs under an Erdős–Rényi model (Erdős & Rényi, 1959). We sample weights independently from the standard normal distribution and transform them with the softplus function.
- **Hierarchical-Weight (HW-) Tree**: 1,000 bifurcating trees with hierarchically sampled edge weights: for each tree $T_i$, sample $\mu_i \sim \mathcal{U}(7, 13)$ from the uniform distribution, then $w_{ik} \sim \Gamma(\mu_i^2, \mu_i^{-1})$ from the gamma distribution. This yields a global weight distribution with mean 10 and standard deviation 2, and within-tree standard deviation of 1.
- **3D Point Cloud**: 41 graphs of household objects (Neumann et al., 2013). Weights from the 3D Point Cloud graphs represent the Euclidean distance between the two nodes in each edge.
- **Lobster**: 1,000 graphs that are path graphs with edges appended at most two edges away from the backbone. We sample weights independently from the Beta distribution as $w_k \sim \text{Beta}(5, 15)$.
- **Path-Threshold (PT-) Tree**: 100 trees where weights and topology are coupled. We sample weights $w_k \sim \mathcal{U}(0.5, 1.5)$ independently. Starting at the root node, we add edges until the sum of outgoing edge weights exceeds a threshold of 4. For each new child, add edges recursively using the same rule: add children until the total weight of all new edges - plus the path length from the root to the current node - exceeds 4. This process continues until all root-to-leaf path sums exceed 4.
- **Yeast**: 100 phylogenetic trees representing the evolutionary history of 154 *Saccharomyces Cerevisiae* strains inferred from DNA sequence alignments. Edge weights denote evolutionary time along each branch from ancestor to descendant (Hassler et al., 2022).

### 4.1   Results

**Weighted Graph Distributions**   BiGG-E consistently outperforms competing models on topological metrics (Table 1). With the HW-tree and lobster graphs, it consistently achieves the best or most competitive MMDs. For the Erdős–Rényi graphs, all BiGG extensions are competitive with the baseline, where the models successfully capture a known probability distribution. All models perform equally well, which is expected given the graphs are fully independent. On the complex 3D point clouds, BiGG+GCN performs best on topology, with BiGG-E remaining competitive with respect to the clustering (0.188 vs 0.179) and spectral ($8.19 \times 10^{-3}$ vs $7.33 \times 10^{-3}$) MMDs. BiGG-E also achieves the lowest error rates for HW-tree and lobster graphs (2.5% and 0.5%, respectively). Finally, BiGG-MLP exhibits substantial degradation in topological quality across graph datasets, performing orders of magnitude worse on HW-tree and 3D point cloud graphs, and showing moderate degradation on lobster graphs.

Table 1: Topological Accuracy Measures. The MMD metrics use the test functions from the set {Degree, Cluster, Orbit, Unweighted Laplacian (Spec)}. For the MMD metrics, smaller values are better. OOM indicates out of memory. Error is reported as the proportion of non-tree or -lobster graphs. Similar or better topological accuracy compared to the original BiGG (BiGG+GCN) shows that modeling of weights does not worsen BiGG-E's topological performance. Results are given as Mean ± SD over five runs. Boldfaced entries represent the best mean value. $< 0.01$ indicates near-zero values and statistical ties on given scale. Em-dash indicates no run was performed. Rank is best model compared with {Mean-SD, Mean, Mean+SD}}.

| Datasets | | | Methods | | | | |
|---|---|---|---|---|---|---|---|
| | | | BiGG-E | Adj-LSTM | BiGG-MLP | BiGG+GCN | Erdős–Rényi |
| Erdős–Rényi $|V|_{max} = 749\ (499)$ $|E|_{max} = 2846\ (1349)$ | Deg. | $(\times 10^{-3})$ | **2.52** ± 0.90 | 30.3 ± 8.6 | 9.56 ± 2.06 | 4.76 ± 1.38 | 2.96 ± 0.65 |
| | Clus. | $(\times 10^{-2})$ | 1.68 ± 1.02 | 3.64 ± 1.18 | 1.66 ± 0.94 | 1.58 ± 0.91 | **1.35** ± 0.38 |
| | Orbit | $(\times 10^{-2})$ | **6.40** ± 0.94 | 10.2 ± 2.5 | 7.98 ± 0.88 | 8.55 ± 2.54 | 6.73 ± 0.39 |
| | Spec. | $(\times 10^{-3})$ | **2.21** ± 0.05 | 24.4 ± 5.9 | 7.38 ± 1.98 | 3.15 ± 0.39 | 2.75 ± 0.51 |
| | Rank | | **1.67** ± 1.15 | 5.00 ± 0.00 | 3.58 ± 0.51 | 2.83 ± 0.72 | 1.92 ± 0.90 |
| HW-Tree $|V|_{max} = 199\ (199)$ $|E|_{max} = 198\ (198)$ | Deg. | $(\times 10^{-4})$ | **0.02** ± 0.01 | 14.1 ± 3.7 | 2.88 ± 0.38 | 0.08 ± 0.05 | 2698.1 ± 29.6 |
| | Spec. | $(\times 10^{-4})$ | 5.75 ± 0.98 | 18.7 ± 1.6 | 20.0 ± 4.1 | **5.71** ± 0.63 | 812.2 ± 13.5 |
| | Orbit | $(\times 10^{-4})$ | **< 0.01** | 0.72 ± 0.05 | 0.12 ± 0.03 | **< 0.01** | 381.6 ± 15.0 |
| | Error | (%) | **2.30** ± 0.30 | 30.20 ± 3.33 | 82.80 ± 3.77 | 5.80 ± 0.97 | 100.00 ± 0.00 |
| | Rank | | **1.29** ± 0.40 | 3.58 ± 0.51 | 3.42 ± 0.51 | 1.71 ± 0.40 | 5.00 ± 0.00 |
| 3D Point Cloud $|V|_{max} = 5022\ (1375)$ $|E|_{max} = 10794\ (3061)$ | Deg. | $(\times 10^{-3})$ | 1.50 ± 1.25 | OOM | 26.1 ± 0.1 | **0.11** ± 0.09 | 534.2 ± 2.81 |
| | Clus. | $(\times 10^{-1})$ | 1.88 ± 0.16 | OOM | 5.77 ± 0.80 | **1.74** ± 0.01 | 11.5 ± 0.04 |
| | Orbit | $(\times 10^{-2})$ | 1.75 ± 1.31 | OOM | 17.9 ± 5.0 | **0.51** ± 0.57 | 102.6 ± 1.53 |
| | Spec. | $(\times 10^{-3})$ | 8.19 ± 0.83 | OOM | 27.5 ± 2.0 | **7.33** ± 0.20 | 254.6 ± 9.02 |
| | Rank | | 1.92 ± 0.29 | 5.00 ± 0.00 | 3.00 ± 0.00 | **1.08** ± 0.29 | 4.00 ± 0.00 |
| Lobster $|V|_{max} = 100\ (55)$ $|E|_{max} = 99\ (54)$ | Deg. | $(\times 10^{-3})$ | **0.32** ± 0.17 | 0.54 ± 0.35 | 2.36 ± 0.45 | 1.17 ± 0.70 | 249.1 ± 0.7 |
| | Clus. | $(\times 10^{-3})$ | **0.00** ± 0.00 | 0.11 ± 0.15 | **0.00** ± 0.00 | **0.00** ± 0.00 | 85.8 ± 10.6 |
| | Spec. | $(\times 10^{-3})$ | **2.80** ± 0.24 | 3.09 ± 0.47 | 4.19 ± 0.69 | 3.27 ± 0.48 | 202.1 ± 3.12 |
| | Orbit | $(\times 10^{-3})$ | **3.33** ± 0.57 | 3.75 ± 0.85 | 29.8 ± 3.1 | 12.33 ± 4.36 | 156.2 ± 2.18 |
| | Error | (%) | **0.90** ± 0.50 | 21.70 ± 4.19 | 31.70 ± 1.96 | 17.00 ± 1.12 | 100.00 ± 0.00 |
| | Rank | | **1.23** ± 0.50 | 2.50 ± 0.73 | 3.63 ± 0.77 | 2.63 ± 0.48 | 5.00 ± 0.00 |
| PT-Tree $|V|_{max} = 452\ (193)$ $|E|_{max} = 451\ (192)$ | Deg. | $(\times 10^{-4})$ | **0.22** ± 0.05 | 152.6 ± 10.3 | 62.2 ± 15.0 | 4.05 ± 2.80 | 602.8 ± 48.3 |
| | Spec. | $(\times 10^{-3})$ | **4.92** ± 0.84 | 261.1 ± 7.4 | 65.7 ± 7.4 | 9.74 ± 2.87 | 229.7 ± 5.5 |
| | Orbit | $(\times 10^{-4})$ | **0.36** ± 0.13 | 349.6 ± 15.1 | 44.7 ± 6.0 | 4.64 ± 2.60 | 102.2 ± 15.0 |
| | Error | (%) | **0.00** ± 0.00 | 100.00 ± 0.00 | **0.00** ± 0.00 | **0.00** ± 0.00 | 100.00 ± 0.00 |
| | Rank | | **1.25** ± 0.45 | 4.38 ± 0.43 | 2.75 ± 0.45 | 2.00 ± 0.00 | 4.64 ± 0.54 |
| Yeast $|V|_{max} = 307\ (307)$ $|E|_{max} = 306\ (306)$ | Deg. | $(\times 10^{-4})$ | **0.33** ± 0.75 | 1165.1 ± 62.6 | 231.8 ± 6.8 | 0.34 ± 0.77 | 2712.4 ± 38.4 |
| | Spec. | $(\times 10^{-3})$ | 0.46 ± 0.44 | 59.2 ± 1.3 | 27.6 ± 5.3 | **0.37** ± 0.34 | 133.4 ± 3.08 |
| | Orbit | $(\times 10^{-5})$ | **0.24** ± 0.54 | 743.6 ± 210.0 | 219.8 ± 95.6 | 0.47 ± 1.06 | 4183.0 ± 262.8 |
| | Error | (%) | **1.00** ± 2.24 | 100.00 ± 0.00 | 87.00 ± 4.47 | **1.00** ± 2.24 | 100.00 ± 0.00 |
| | Rank | | **1.38** ± 0.38 | 4.13 ± 0.23 | 3.00 ± 0.00 | 1.63 ± 0.38 | 4.88 ± 0.23 |
| **Global Rank** | | | **1.45** ± 0.63 | 4.03 ± 1.00 | 3.25 ± 0.58 | 2.01 ± 0.74 | 4.27 ± 1.16 |

Table 2: Edge-weight Accuracy Measures. The MMD metrics use the test functions from the set {Weighted Laplacian (Spec) and Weights}. For the MMD metrics, smaller values are better. OOM indicates out of memory. Marginal means and standard deviations of edge weights are given as $\mu_w$ and $\sigma_w$ for underlying edge weight distributions, and $\bar{w}$ and $s_w$ when computed from sampled graphs. For ER Graphs, edge-weights are on the underlying normal distribution scale. For HW-tree graphs, $s_T$ represents the average standard deviation of edge weights sampled per HW-tree. Results are given as Mean ± SD over five runs. Boldfaced entries represent the best mean value. Em-dash indicates no run was performed. Rank is best model compared with {Mean-SD, Mean, Mean+SD}

| Datasets | | | Methods | | | | |
|---|---|---|---|---|---|---|---|
| | | | BiGG-E | Adj-LSTM | BiGG-MLP | BiGG+GCN | Erdős–Rényi |
| Erdős–Rényi $\mu_w = 0.0$ $\sigma_w = 1.0$ | $\bar{w}$ | $(\times 10^{-3})$ | $5.12$ ±3.13 | $19.4$ ±4.0 | $3.31$ ±4.12 | $-3.84$ ±7.07 | $\mathbf{2.96}$ ±0.65 |
| | $s_w$ | | $1.01$ ±0.01 | $0.98$ ±0.00 | $1.01$ ±0.01 | $\mathbf{1.00}$ ±0.00 | $\mathbf{1.00}$ ±0.00 |
| | Spec. | $(\times 10^{-3})$ | $3.79$ ±0.43 | $27.1$ ±7.3 | $7.38$ ±1.98 | $4.12$ ±0.70 | $\mathbf{2.80}$ ±0.38 |
| | MMDWT | $(\times 10^{-3})$ | $\mathbf{1.27}$ ±0.66 | $38.6$ ±12.2 | $1.97$ ±1.18 | $1.85$ ±0.51 | $1.76$ ±1.12 |
| | Rank | | $2.42$ ±1.18 | $4.92$ ±0.29 | $3.25$ ±0.99 | $2.63$ ±0.96 | $\mathbf{1.79}$ ±0.75 |
| HW-Tree $\mu_w = 10,\ \sigma_w = 2$ $\sigma_T = 1$ | $\bar{w}$ | | $10.13$ ±0.12 | $\mathbf{9.97}$ ±0.07 | $10.13$ ±0.14 | $10.03$ ±0.10 | $10.04$ ±0.01 |
| | $s_w$ | | $1.86$ ±0.02 | $\mathbf{1.98}$ ±0.04 | $1.93$ ±0.02 | $1.96$ ±0.03 | $1.96$ ±0.00 |
| | $s_T$ | | $1.01$ ±0.01 | $1.32$ ±0.04 | $0.98$ ±0.00 | $1.96$ ±0.03 | $1.96$ ±0.00 |
| | Spec. | $(\times 10^{-3})$ | $\mathbf{0.63}$ ±0.07 | $2.83$ ±0.54 | $1.33$ ±0.14 | $2.09$ ±0.12 | $109.3$ ±1.8 |
| | MMDWT | $(\times 10^{-2})$ | $0.85$ ±0.33 | $\mathbf{0.33}$ ±0.09 | $0.71$ ±0.50 | $21.6$ ±0.1 | $21.5$ ±0.1 |
| | Rank | | $2.67$ ±1.49 | $\mathbf{2.50}$ ±1.32 | $2.67$ ±1.30 | $3.50$ ±1.28 | $3.67$ ±1.22 |
| 3D Point Cloud $\mu_w \approx 0.411$ $\sigma_w \approx 0.096$ | $\bar{w}$ | $(\times 10^{-1})$ | $\mathbf{4.18}$ ±0.06 | OOM | $4.28$ ±0.02 | $4.56$ ±0.06 | $4.20$ ±0.01 |
| | $s_w$ | $(\times 10^{-1})$ | $\mathbf{0.96}$ ±0.01 | OOM | $1.04$ ±0.01 | $7.37$ ±1.14 | $0.98$ ±0.00 |
| | Spec. | $(\times 10^{-3})$ | $7.77$ ±0.90 | OOM | $23.1$ ±0.64 | $8.52$ ±0.92 | $287.6$ ±5.4 |
| | MMDWT | $(\times 10^{-2})$ | $\mathbf{0.27}$ ±0.19 | OOM | $5.15$ ±0.71 | $18.3$ ±2.2 | $2.00$ ±0.78 |
| | Wtd. Deg. | $(\times 10^{-3})$ | $\mathbf{7.31}$ ±5.83 | OOM | $87.6$ ±2.14 | $9.67$ ±1.29 | $455.8$ ±2.4 |
| | Rank | | $\mathbf{1.13}$ ±0.35 | $5.00$ ±0.00 | $3.00$ ±0.00 | $3.13$ ±1.12 | $2.73$ ±1.10 |
| Lobster $\mu_w = 0.25$ $\sigma_w \approx 0.095$ | $\bar{w}$ | $(\times 10^{-1})$ | $2.49$ ±0.01 | $2.50$ ±0.00 | $2.49$ ±0.01 | $\mathbf{2.50}$ ±0.01 | $2.49$ ±0.00 |
| | $s_w$ | $(\times 10^{-1})$ | $\mathbf{0.94}$ ±0.01 | $1.03$ ±0.01 | $\mathbf{0.94}$ ±0.01 | $\mathbf{0.94}$ ±0.01 | $\mathbf{0.94}$ ±0.01 |
| | Spec. | $(\times 10^{-3})$ | $\mathbf{2.44}$ ±0.16 | $3.09$ ±0.47 | $4.19$ ±0.39 | $3.27$ ±0.48 | $271.7$ ±2.4 |
| | MMDWT | $(\times 10^{-3})$ | $1.33$ ±0.45 | $6.36$ ±1.54 | $\mathbf{1.21}$ ±0.36 | $1.60$ ±0.30 | $9.15$ ±1.71 |
| | Rank | | $\mathbf{1.92}$ ±0.67 | $3.79$ ±1.23 | $2.42$ ±1.16 | $3.21$ ±0.78 | $3.67$ ±1.50 |
| PT-Tree $\mu_w = 1.00$ $\sigma_w \approx 0.29$ | $\bar{w}$ | | $\mathbf{1.00}$ ±0.00 | $1.03$ ±0.01 | $0.99$ ±0.01 | $\mathbf{1.00}$ ±0.00 | $\mathbf{1.00}$ ±0.00 |
| | $s_w$ | | $\mathbf{0.29}$ ±0.00 | $0.30$ ±0.00 | $0.28$ ±0.00 | $\mathbf{0.29}$ ±0.00 | $\mathbf{0.29}$ ±0.00 |
| | Spec. | $(\times 10^{-2})$ | $1.35$ ±0.23 | $18.3$ ±0.5 | $2.76$ ±0.28 | $1.94$ ±0.33 | $21.2$ ±0.8 |
| | MMDWT | $(\times 10^{-3})$ | $4.03$ ±0.20 | $18.9$ ±2.2 | $7.57$ ±1.62 | $22.6$ ±2.8 | $\mathbf{0.54}$ ±0.38 |
| | Wtd Deg. | $(\times 10^{-3})$ | $\mathbf{0.38}$ ±0.08 | $8.92$ ±0.78 | $2.38$ ±0.36 | $4.12$ ±0.11 | $74.0$ ±8.1 |
| | Rank | | $\mathbf{1.63}$ ±0.55 | $4.37$ ±0.48 | $3.13$ ±0.81 | $2.83$ ±1.19 | $3.03$ ±1.72 |
| Yeast $\mu_w \approx 3.63 \times 10^{-4}$ $\sigma_w \approx 3.68 \times 10^{-4}$ | $\bar{w}$ | $(\times 10^{-4})$ | $\mathbf{3.65}$ ±0.01 | $6.22$ ±0.08 | $\mathbf{3.65}$ ±0.02 | $3.47$ ±0.02 | $3.66$ ±0.05 |
| | $s_w$ | $(\times 10^{-4})$ | $3.71$ ±0.03 | $9.50$ ±0.26 | $3.87$ ±0.22 | $\mathbf{3.67}$ ±0.06 | $3.79$ ±0.06 |
| | Spec. | $(\times 10^{-3})$ | $3.28$ ±0.85 | $98.6$ ±4.2 | $18.6$ ±2.9 | $23.7$ ±1.6 | $133.4$ ±3.1 |
| | MMDWT | $(\times 10^{-3})$ | $\mathbf{0.56}$ ±0.10 | $31.8$ ±9.3 | $1.36$ ±0.63 | $34.6$ ±3.7 | $5.59$ ±1.93 |
| | Wtd Deg. | $(\times 10^{-4})$ | $\mathbf{0.22}$ ±0.06 | $358.0$ ±19.1 | $14.4$ ±4.0 | $15.3$ ±1.5 | $211.6$ ±14.3 |
| | Rank | | $\mathbf{1.23}$ ±0.42 | $4.67$ ±0.49 | $2.23$ ±0.82 | $3.27$ ±1.22 | $3.60$ ±0.83 |
| **Global Rank** | | | $\mathbf{1.81}$ ±1.07 | $4.20$ ±1.17 | $2.78$ ±0.98 | $3.11$ ±1.12 | $3.11$ ±1.36 |

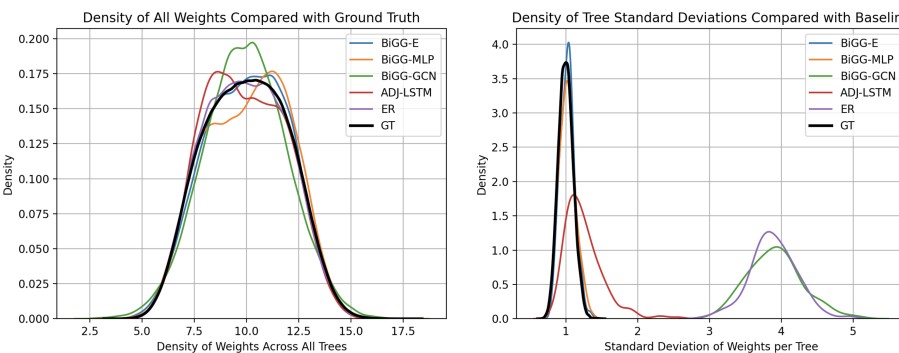

Figure 4: Distribution of weights globally (left) and of standard deviations per HW-tree (right).

**Joint Modeling**  BiGG-E outperforms all comparisons in jointly modeling weighted graphs, particularly on the 3D point cloud, PT-tree, and yeast graphs (Table 2) . Although BiGG+GCN achieves a lower unweighted degree distribution MMD for 3D point clouds, BiGG-E attains a lower weighted degree distribution MMD ($7.31 \times 10^{-3}$ vs $9.67 \times 10^{-3}$). It also matches with BiGG+GCN on the weighted Laplacian MMD ($2.44 \times 10^{-3}$ vs $3.27 \times 10^{-3}$), where BiGG-E's more realistic edge weights improve joint modeling.

In the PT-tree dataset, where topology and edge weights are sampled jointly, BiGG-E achieves the best performance across all metrics by significant margins. Although the Erdős–Rényi baseline performs well on the marginal distribution of weights, which are independently and uniformly distributed, BiGG-E still closely matches this distribution despite using a softplus-parameterized distribution, while also capturing the structural dependencies exhibited in the PT-trees that the baseline cannot model. On yeast graphs, BiGG-E is comparable to BiGG+GCN on topology-only metrics but consistently leads once edge weights are included. More broadly, BiGG+GCN leverages its BiGG component on unweighted topologies that depend little on the edge weights, while BiGG-E outperforms when topology is edge weight-dependent, suggesting its joint modeling captures weighted graph structures more effectively.

In contrast to BiGG-E's stable edge weight learning, the two-stage BiGG+GCN pipeline is highly sensitive to errors in the generated topology of the graph, highlighted by the heavily inflated variance of edge weights produced in the 3D point clouds. Uncertainty in the graph topology propagates to the GCN, leading to instability in the sampled set of edge weights. BiGG-E's joint modeling scheme leads to more stable and realistic weighted graph generation by allowing uncertainty to flow in both directions: uncertainty in topology of the underlying graphs propagates to uncertainty in the edge weight distribution, and vice versa.

**Scalability**  Table 3 shows that all BiGG extensions scale well to larger graphs, with BiGG-E performing best. Adj-LSTM's performance deteriorates rapidly even on the moderately sized Erdős–Rényi graphs and becomes computationally infeasible for the 3D point clouds, as observed in Table 1. Furthermore, Table 3 shows Adj-LSTM fails in scaling to graphs beyond even 500 nodes, where runtime becomes prohibitively slow. On the other hand, all BiGG extensions scale to graphs with thousands of nodes. In Figure 3, we empirically demonstrate that all BiGG extensions remain efficient for large graph generation, with training time scaling as $\mathcal{O}(\log n)$ and sampling time as $\mathcal{O}((n+m)\log n)$, while the training times for Adj-LSTM and SparseDiff increase rapidly with respect to graph size and quickly become impractical. Although BiGG-E's incorporation of a separate weight state slightly increases training and sampling time, the overhead is minimal and justifiable with the superior generative quality.

Notably, separating the topology and weight states allows BiGG-E to improve memory efficiency. The original BiGG model applies bits compression to summarize the node intervals in $\mathcal{T}_u$ with binary vectors, significantly reducing neural network computation and memory usage (Dai et al., 2020). While this is not feasible in BiGG-MLP from entangling edge weights with the topology state, incorporating bits compression into BiGG-E's topology state is trivial. Moreover, BiGG-E offers flexibility in the dimensionality of each state: we use embedding dimensions of 256 and 32 for the topology and weight states, respectively. This re-

Table 3: Model Scaling on HW-Trees of Varying Size. Weighted Laplacian MMD is reported. Graphs are weighted HW-trees of increasing size. Lower is better. OOM indicates Out of Memory.

| Model | 100 | 0.5K | 1K | 2K | 5K | 10K | 15K |
|---|---|---|---|---|---|---|---|
| Erdős–Rényi | 0.073 | 0.103 | 0.114 | 0.122 | 0.128 | 0.131 | 0.134 |
| BiGG-E | $2.77_{\times 10^{-3}}$ | $\mathbf{2.65_{\times 10^{-3}}}$ | $\mathbf{1.31_{\times 10^{-3}}}$ | $\mathbf{5.60_{\times 10^{-4}}}$ | $\mathbf{3.13_{\times 10^{-4}}}$ | $\mathbf{6.00_{\times 10^{-4}}}$ | $\mathbf{5.12_{\times 10^{-4}}}$ |
| BiGG-MLP | $\mathbf{1.36_{\times 10^{-3}}}$ | $4.82_{\times 10^{-3}}$ | $3.18_{\times 10^{-3}}$ | $2.04_{\times 10^{-3}}$ | $1.44_{\times 10^{-3}}$ | $6.32_{\times 10^{-4}}$ | $5.47_{\times 10^{-4}}$ |
| Adj-LSTM | $6.11_{\times 10^{-3}}$ | $9.27_{\times 10^{-3}}$ | OOM | OOM | OOM | OOM | OOM |
| BiGG+GCN | $3.67_{\times 10^{-3}}$ | $6.01_{\times 10^{-3}}$ | $2.78_{\times 10^{-3}}$ | $1.81_{\times 10^{-3}}$ | $2.96_{\times 10^{-3}}$ | $4.31_{\times 10^{-3}}$ | $1.74_{\times 10^{-3}}$ |

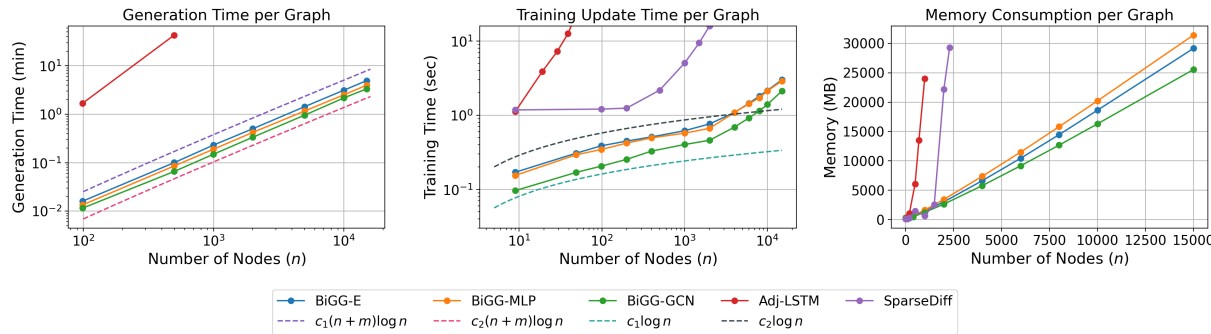

Figure 5: Model Scalability. Sampling time per weighted graph (left); training time per weighted graph (middle); and memory consumption per graph during training (right).

duces computational overhead and helps prevent overfitting on weights, where we empirically observe a 20% reduction in BiGG-E's memory consumption. BiGG-MLP, however, uses dense MLPs to output edge embeddings that must match the topology state's embedding dimension of 256, contributing to unstable training and increased memory use. As a result, BiGG-E matches BiGG-MLP's training time while consuming less memory and producing higher-quality weighted graphs.

## 5 Conclusion and Future Work

We introduce an autoregressive model that learns complex joint distributions over graphs with edge weights. While Adj-LSTM and both BiGG extensions learn from distributions over smaller graphs, BiGG-E scales best to graphs with thousands of nodes while maintaining strong performance learning joint distributions over graph topologies and edge weights. Future work consists of further improving BiGG-E, especially with respect to memory consumption. In addition, we can further explore the benefits of joint modeling edge weights and topologies by learning joint distributions over topologies and vectors of edge and node attributes, learning conditional distributions over these given node- or edge- related data. Finally and thanks to its scalability, BiGG-E may also be useful within larger regression models featuring network outcomes.

## 6 Acknowledgments

We would like to extend our thanks to Dr. Hanjun Dai for his help in extending the code base of BiGG to use BiGG-MLP.

Richard Williams is supported by NSF grant DMS 2236854. Andrew Holbrook is supported by the NSF (DMS 2236854, DMS 2152774) and the NIH (K25 AI153816). This work was made possible by the support of the Cure Alzheimer's Fund and the Kavli Foundation.

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

# A  Appendix

## A.1  Probability Calculation of a Weighted Adjacency Matrix

Here, we derive the probability of observing $\mathbf{W}$ through (1) all entries in a row-wise manner and (2) its weighted edge set. First, we define $p_{\boldsymbol{\theta}}(\mathbf{A})$ over unweighted graphs and then extend these probabilities to probabilities $p_{\boldsymbol{\theta}}(\mathbf{W})$ over weighted graphs. One method in estimating $p_{\boldsymbol{\theta}}(\mathbf{A})$ is to directly generate all of the lower half of $\mathbf{A}$ in a row-wise fashion. Letting $A_{ij}$ represent the $(i,j)$-th element of $\mathbf{A}$, we have that

$$p_{\boldsymbol{\theta}}(\mathbf{A}) = \prod_{i=1}^{n}\prod_{j=1}^{i-1} p_{\boldsymbol{\theta}}(A_{ij}|\{A_{kl}\}). \tag{11}$$

where $\{A_{kl}\}$ is the set of all entries of $\mathbf{A}$ that come before $A_{ij}$ in the row generation process.

Models such as Li et al. (2018) and You et al. (2018) use this direct factorization of the probability of all entries of $\mathbf{A}$, leading to the computational bottleneck of $\mathcal{O}(n^2)$ for a graph with $n$ nodes. BiGG (Dai et al., 2020), on the other hand, leverages the sparsity of many real-world graphs and directly generates the edge-set of $\mathbf{A}$, leading to the probability

$$p_{\boldsymbol{\theta}}(\mathbf{A}) = \prod_{i=1}^{m} p_{\boldsymbol{\theta}}(e_k|\{e_{l;l<k}\}). \tag{12}$$

where $e_k$ is the $i^{\text{th}}$ ordered edge in $G$.

Given the choice of parameterization outlined in Section 3.1, we modify equation 11 and equation 12 as follows: given an edge exists between nodes $v_i$ and $v_j$, sample a corresponding weight $w$ from a conditional distribution parameterized by $p_{\boldsymbol{\theta}}(w|e)$.

For readability, assume entries $W_{ij}$ are conditioned on all prior entries; that is, $p_{\boldsymbol{\theta}}(W_{ij}) \equiv p_{\boldsymbol{\theta}}(W_{ij}|\{W_{kl}\})$, where $\{W_{kl}\}$ is the set of entries of $\mathbf{W}$ that come prior to entry $W_{ij}$ when traversing the lower half of $\mathbf{A}$ in a row-by-row manner.

For each entry $W_{ij}$, we must decide if an edge connects nodes $v_i$ and $v_j$ by sampling $e_{ij} \sim \text{Bernoulli}(p_{ij})$, where $p_{ij}$ is a function of $\boldsymbol{\theta}$, and if so, sample a non-negative weight $w_{ij}$ with probability $p_{\boldsymbol{\theta}}(w_{ij}|e_{ij}=1)$. Thus, we note that the probabilities of a particular entry $W_{ij}$ being 0 or weight $w_{ij}$ are given by

$$p_{\boldsymbol{\theta}}(W_{ij}=0) = p_{\boldsymbol{\theta}}(e_{ij}=0) = 1 - p_{ij}$$

and

$$p_{\boldsymbol{\theta}}(W_{ij}=w_{ij}) = p_{\boldsymbol{\theta}}(e_{ij}=1)p_{\boldsymbol{\theta}}(w_{ij}|e_{ij}=1) = p_{ij}p_{\boldsymbol{\theta}}(w_{ij}|e_{ij}=1).$$

We may succinctly represent the probability of a $W_{ij}$ as

$$p_{\boldsymbol{\theta}}(W_{ij}=w_{ij}) = (1-p_{ij})^{1-e_{ij}}\left[p_{ij}p_{\boldsymbol{\theta}}(w_{ij}|e_{ij})\right]^{e_{ij}}.$$

Finally, we define the probability of $\mathbf{W}$ as

$$p_{\boldsymbol{\theta}}(\mathbf{W}) = \prod_{i=1}^{n}\prod_{j=1}^{i-1}(1-p_{ij})^{1-e_{ij}}\left[p_{ij}p_{\boldsymbol{\theta}}(w_{ij}|e_{ij})\right]^{e_{ij}}.$$

Next, we modify equation 11 to obtain the probability of observing $\mathbf{W}$ over all entries as

$$p_{\boldsymbol{\theta}}(\mathbf{W}) = \prod_{i=1}^{n}\prod_{j=1}^{i-1} p_{\boldsymbol{\theta}}(W(v_i,v_j)|\{W(v_k,v_l)\}).$$

On the other hand, because no weight is sampled for non-edge connections, the probability of observing the edge set of $\mathbf{W}$ becomes

$$p_{\boldsymbol{\theta}}(\mathbf{W}) = \prod_{i=1}^{m} p_{\boldsymbol{\theta}}(e_k, w_k | \{(w_i, e_i)\}_{i<k}),$$

where we factorize $p_{\boldsymbol{\theta}}(e_k, w_k | \cdot) = p_{\boldsymbol{\theta}}(e_k | \cdot) p_{\boldsymbol{\theta}}(w_k | e_k, \cdot)$ using equation 2.

Finally, we compute our likelihood objective functions. First, we compute $\mathcal{L}(\boldsymbol{\theta}; \mathbf{W})$ from equation 4, which is used to train Adj-LSTM. The training objective is the log-likelihood over all entries $\mathbf{W}$, summed over the terms

$$\ell_{ij}\big(\boldsymbol{\theta}; (e_{ij}, w_{ij})\big) = (1 - e_{ij}) \log(1 - p_{ij}) + e_{ij} \log p_{ij} + e_{ij} \log p_{\boldsymbol{\theta}}(w_{ij}),$$

where $e_{ij} \log p_{\boldsymbol{\theta}}(w_{ij} | e_{ij}) = 0$ if $e_{ij} = 0$, and we substitute Equation 3 into the expression $p_{\boldsymbol{\theta}}(w_{ij} | e_{ij})$ when $e_{ij} = 1$. This yields the training objective for Adj-LSTM as

$$\mathcal{L}(\boldsymbol{\theta}; \mathbf{W}) = \log \prod_{i=1}^{n} \prod_{j=1}^{i-1} p_{\boldsymbol{\theta}}(W_{ij}) = \sum_{i=1}^{n} \sum_{j=1}^{i-1} \ell_{ij}(\boldsymbol{\theta}; (e_{ij}, w_{ij})). \tag{13}$$

Last, from Equation 5, the objective function for all BiGG extensions is the log likelihood over the weighted edge set,

$$\mathcal{L}(\boldsymbol{\theta}; \mathbf{W}) = \sum_{k=1}^{m} \log p_{\boldsymbol{\theta}}(e_k | \{(e_l, w_l)_{:l<k}\}) + \log p_{\boldsymbol{\theta}}(w_k | e_k, \{(e_l, w_l)_{:l<k}\}), \tag{14}$$

where we substitute Equation 3 into the expression $p_{\boldsymbol{\theta}}(w_k | e_k, \{(e_l, w_l)_{:l<k}\})$.

## A.2 Model Architecture

Autoregressive models are popular sequential models designed to capture dependencies with prior observations. A classical example in statistics is the $AR(\rho)$ model, where each observation $X_t$ is expressed as a linear combination of the previous $\rho$ values in the series: $X_t = \sum_{i=1}^{\rho} \varphi_i X_{t-i} + \epsilon_t$. Deep autoregressive models extend this idea by introducing nonlinear function approximators, such as neural networks, to better model dependencies in sequential data. In this section, we describe in detail the primary neural network architectures used in building each autoregressive model.

**LSTM Architecture**  The neural network architecture across all models utilizes LSTM cells, which are recurrent neural networks well-suited for modeling nonlinearities in sequential data in an autoregressive manner (Hochreiter & Schmidhuber, 1997). An LSTM maintains a state, represented as a tuple $(\mathbf{h}_t, \mathbf{c}_t)$ where $\mathbf{h}_t, \mathbf{c}_t \in \mathbb{R}^d$ are the hidden and cell states, respectively, at every time step $t$. Here, $\mathbf{h}_t$ captures the recent memory of the sequence, while $\mathbf{c}_t$ encodes longer-term memory. Given an input $\mathbf{x}_t$ at time step $t$ and the previous state $(\mathbf{h}_{t-1}, \mathbf{c}_{t-1})$, the LSTM updates the state by computing $(\mathbf{h}_t, \mathbf{c}_t)$ as follows (Sak et al., 2014):

- $\mathbf{i}_t = \sigma(\mathbf{W}_x^{(i)} \mathbf{x}_t + \mathbf{W}_h^{(i)} \mathbf{h}_{t-1} + \mathbf{b}^{(i)})$
- $\mathbf{f}_t = \sigma(\mathbf{W}_x^{(f)} \mathbf{x}_t + \mathbf{W}_h^{(f)} \mathbf{h}_{t-1} + \mathbf{b}^{(f)})$
- $\mathbf{g}_t = \tanh(\mathbf{W}_x^{(g)} \mathbf{x}_t + \mathbf{W}_h^{(g)} \mathbf{h}_{t-1} + \mathbf{b}^{(g)})$
- $\mathbf{o}_t = \sigma(\mathbf{W}_x^{(o)} \mathbf{x}_t + \mathbf{W}_h^{(o)} \mathbf{h}_{t-1} + \mathbf{b}^{(o)})$
- $\mathbf{c}_t = \mathbf{f}_t \odot \mathbf{c}_{t-1} + \mathbf{i}_t \odot \mathbf{g}_t$
- $\mathbf{h}_t = \mathbf{o}_t \odot \tanh(\mathbf{c}_t)$

where $\sigma$ is the sigmoid function, $\odot$ is the Hadamard product, and all weight matrices and bias terms are trainable parameters. $\mathbf{i}_t$, $\mathbf{f}_t$, $\mathbf{g}_t$, and $\mathbf{o}_t$ represent the input, forget, cell, and output gates, respectively. We will also use the notation $(\mathbf{h}_{t+1}, \mathbf{c}_{t+1}) = \mathrm{LSTM}(\mathbf{x}_t, (\mathbf{h}_t, \mathbf{c}_t))$ to represent these computations. For ease of readability, we suppress subscript $t$ moving forward.

**MLP** In the graph generative modeling context, an LSTM cell serves to maintain a history of the graph generated by the model at each time step. At each time step, the model must make a prediction—either regarding topology (i.e., the probability of whether an edge exists) or regarding edge weight parameters (e.g., producing the mean and variance of a softplus-transformed normal distribution). To compute these predictions, the hidden state $\mathbf{h}$ is passed through an MLP. An MLP is a feedforward neural network composed of an input layer, one or more hidden layers, and an output layer, where it captures nonlinear patterns in the data through activation functions applied between layers (Rumelhart et al., 1986). In our architecture, we use the exponential linear unit (ELU) as the activation function:

$$\mathrm{ELU}(x) = \begin{cases} x & x \geq 0 \\ \alpha(e^x - 1) & x < 0 \end{cases}$$

where $\alpha$ is a tuned parameter. In our case, the inputs consist of the LSTM hidden state of the model, and the outputs are probabilities of edge connections, as well as the mean and variance components $\mu_k$ and $\sigma_k^2$ introduced in equation 3. For notational simplicity, let $\mathbf{h}_G$ denote the current hidden state $\mathbf{h}$ summarizing the graph generated thus far. To compute edge existence probabilities, we set $p = \sigma(f_p(\mathbf{h}_G))$, where $f_p$ is the MLP that outputs a scalar that is mapped to a probability by the sigmoid function $\sigma$. Similarly, the softplus-transformed normal parameters $\mu_k$ and $\sigma_k^2$ are parameterized with MLPs $f_\mu$ and $f_{\sigma^2}$, respectively: $\mu_k = f_\mu(\mathbf{h}_G)$ and $\sigma_k^2 = \exp(f_{\sigma^2}(\mathbf{h}_G))$.

**Tree-LSTM Cells** Finally, we note that BiGG offers a scalable approach to the typically slow process of training on graph data by leveraging an algorithm with $\mathcal{O}(\log n)$ runtime. This efficiency is achieved with the usage of a Tree-LSTM cell (Tai et al., 2015), a variant of the standard LSTM cell that is designed for tree-structured inputs. Specifically, BiGG and its extensions employ a binary Tree-LSTM cell, where each internal node has exactly two children. Throughout this work, we refer to these simply as Tree-LSTM cells, with the understanding that they are all binary in structure.

Each node in the tree is associated with a hidden state $(\mathbf{h}_j, \mathbf{c}_j)$, where $j$ denotes the index of the corresponding node. Leaf nodes consist of states computed directly from the model, of which no computations are made within the Tree-LSTM cell. For internal nodes, which have two children designated as the left child ($L$) and right child ($R$), the summary state at node $j$ is computed using the states of its children, $jL$ and $jR$, as follows (Tai et al., 2015):

- $\mathbf{i}_j = \sigma(\mathbf{W}_L^{(i)}\mathbf{h}_{jL} + \mathbf{W}_R^{(i)}\mathbf{h}_{jR} + \mathbf{b}^{(i)})$
- $\mathbf{f}_{jL} = \sigma(\mathbf{W}_L^{(fL)}\mathbf{h}_{jL} + \mathbf{W}_R^{(fL)}\mathbf{h}_{jR} + \mathbf{b}^{(f)})$
- $\mathbf{f}_{jR} = \sigma(\mathbf{W}_L^{(fR)}\mathbf{h}_{jL} + \mathbf{W}_R^{(fR)}\mathbf{h}_{jR} + \mathbf{b}^{(f)})$
- $\mathbf{g}_j = \tanh(\mathbf{W}_L^{(g)}\mathbf{h}_{jL} + \mathbf{W}_R^{(g)}\mathbf{h}_{jR} + \mathbf{b}^{(g)})$
- $\mathbf{o}_j = \sigma(\mathbf{W}_L^{(o)}\mathbf{h}_{jL} + \mathbf{W}_R^{(o)}\mathbf{h}_{jR} + \mathbf{b}^{(o)})$
- $\mathbf{c}_j = \mathbf{i}_j \odot \mathbf{g}_j + \mathbf{f}_{jL} \odot \mathbf{c}_{jL} + \mathbf{f}_{jR} \odot \mathbf{c}_{jR}$
- $\mathbf{h}_j = \mathbf{o}_j \odot \tanh(\mathbf{c}_j)$

A few key differences are worth noting. Unlike the standard sequential LSTM, each node in the tree maintains its own hidden state but does not receive an input vector $\mathbf{x}$. Additionally, the Tree-LSTM cell computes distinct forget gates for the left and right children — denoted $\mathbf{f}_{jL}$ and $\mathbf{f}_{jR}$, respectively — which allows the model to attend to different information from each child node (Tai et al., 2015). We use the notation $\mathbf{h}_j = \mathrm{TreeLSTM}(\mathbf{h}_{jL}, \mathbf{h}_{jR})$ to represent the computation at internal node $j$ based on its two children, where the use of the corresponding cell states is implied. In this way, the Tree-LSTM cell represents a mechanism for merging two input states into a single summary state that captures pertinent information from both children.

Finally, we note that all LSTM cells, MLPs, and Tree-LSTM cells are parameterized by neural network parameters $\boldsymbol{\theta}$, which are optimized using the objective functions given in equation 13 (Adj-LSTM) and equation 14 (BiGG models) via gradient descent when training each model on a collection of training graphs.

### A.2.1 Further Details on BiGG (Dai et al., 2020)

BiGG generates an unweighted graph with an algorithm decomposed into two main components, both of which train in $\mathcal{O}(\log n)$ time: (1) row generation, where BiGG generates each row of the lower half of $\mathbf{A}$ using a binary decision tree; and (2) row conditioning, where BiGG deploys a hierarchical data maintenance structure called a Fenwick Tree generate each row conditioned on all prior rows.

### A.2.2 Row Generation

For each node $v_u \in G$, BiGG constructs a binary decision tree $\mathcal{T}_u$, adapted from R-MAT (Chakrabarti et al., 2004), to determine $v_u$'s edge connections with the previously sequenced nodes $v_1, \ldots, v_{u-1}$. Rather than evaluating each potential connection sequentially, which incurs a cost of $\mathcal{O}(n)$ per row of $\mathbf{A}$, BiGG first checks whether any edge exists within the interval $[v_1, v_{u-1}]$. If so, it applies the R-MAT procedure to recursively divide this interval in half and identify the specific nodes connected to $v_u$. By partitioning the interval in halves rather than scanning all possible connections, BiGG reduces the row generation time to $\mathcal{O}(\log n)$.

Let $t \in \mathcal{T}_u$ correspond to the node interval $[v_j, v_k]$ of length $l_t = k - j + 1$, and denote the left-half and right-half of this interval as $\text{lch}(t) = [v_j, v_{j+\lfloor l_t/2 \rfloor}]$ and $\text{rch}(t) = [v_{j+\lfloor l_t/2 \rfloor+1}, v_k]$, respectively. To recursively generate $\mathcal{T}_u$, BiGG uses the following procedure for each $t \in \mathcal{T}_u$. If $t$ is a leaf in the tree — that is, $t$ corresponds to a trivial interval of a single node $v_j$ — an edge forms between $v_j$ and $v_u$. Otherwise, BiGG considers whether an edge exists in $\text{lch}(t)$. If so, the model recurses into $\text{lch}(t)$ until reaching a leaf. After constructing the entire left subtree, the model considers whether an edge exists in $\text{rch}(t)$, conditioned on this left subtree. If so, the model recurses into $\text{rch}(t)$ until reaching a leaf. All leaves in the fully constructed tree thus represent nodes which connect with $v_u$ in the graph.

Each decision about the presence of an edge in the interval corresponding to node $t$ is modeled as a Bernoulli random variable with probability as a function of $\boldsymbol{\theta}$. To make these predictions autoregressive, BiGG maintains two context vectors. The top-down context $\mathbf{h}_u^{\text{top}}(t)$ encodes the history of all previous left-right decisions along the path from the root to the node $t$ and is used to estimate the probability of an edge in the left child, $\text{lch}(t)$. The bottom-up context $\mathbf{h}_u^{\text{bot}}(t)$ captures information from the already generated left subtree rooted at node $t$, and is used to condition the probability of an edge in the right child, $\text{rch}(t)$, based on the structure of $\text{lch}(t)$. Since $\text{rch}(t)$ is constructed only after $\text{lch}(t)$ and its dependencies, BiGG employs a Tree-LSTM cell (Tai et al., 2015) at node $t$ to merge the top-down context with the bottom-up summary of the left subtree prior to generating $\text{rch}(t)$. Hence, the probability of constructing each tree $\mathcal{T}_u$ is

$$p_{\boldsymbol{\theta}}(\mathcal{T}_u) = \prod_{t \in \mathcal{T}_u} p_{\boldsymbol{\theta}}(\text{lch}(t)|\mathbf{h}_u^{top}(t)) \cdot p_{\boldsymbol{\theta}}(\text{rch(t)}|\hat{\mathbf{h}}_u^{top}(t)), \tag{15}$$

where $\hat{\mathbf{h}}_u^{top}(t) = \text{TreeCell}_{\boldsymbol{\theta}}(\mathbf{h}_u^{\text{top}}(t), \mathbf{h}_u^{\text{bot}}(\text{lch}(t)))$, and conditioning $\text{lch}(t)$ and $\text{rch}(t)$ on the top-down and bottom-up context vectors allows BiGG to generate all of $\mathcal{T}_u$ autoregressively. Equation 15 also implies that the probabilities of edges $p_{\boldsymbol{\theta}}(e_k)$ in Equation 1 are modeled as a sequence of left-right edge existence Bernoulli decisions, which are then included in the objective function for all BiGG models. Figure 1 provides an example of constructing $\mathcal{T}_u$ and visualizes the top-down and bottom-up states used to predict left and right edge existence decisions.

### A.2.3 Fenwick Tree

The construction of $\mathcal{T}_u$ in Section A.2.2 enables modeling the dependencies among edge connections within a single row of $\mathbf{A}$. To make BiGG fully autoregressive, however, it must also capture dependencies *between* rows. For this purpose, BiGG incorporates the Fenwick tree (Fenwick, 1994) — a data structure designed to efficiently maintain prefix sums by performing sum operations over an array of length $n$ in $\mathcal{O}(\log n)$ time.

To maintain a history of all previously generated rows in the graph, BiGG modifies the Fenwick tree to store summary representations of each row generated so far. This structure enables BiGG to compute a summary state of the first $u$ rows in $\mathcal{O}(\log u)$ time.

The Fenwick topology tree is structured into $\lfloor \log(u-1) \rfloor + 1$ levels. At the base—level 0—the leaves of the tree represent independent row embeddings for each $\mathcal{T}_u$. These embeddings are constructed using a bottom-up traversal: information starting from the leaf nodes of $\mathcal{T}_u$ propagates upward through the tree, culminating in a root-level summary that captures the overall structure of $\mathcal{T}_u$ and therefore summarizes all node connections with $v_u$. Entries along higher levels of the Fenwick topology tree represent merged state summaries — each combining information from multiple lower-level embeddings to produce a collective summary state across several rows.

Let $\mathbf{g}_j^i = (\mathbf{h}, \mathbf{c})$ denote the hidden and cell states represented by $j$-th node on the $i$-th level $\mathbf{g}_j^i$ of the Fenwick topology tree. Any given non-leaf node $\mathbf{g}_j^i$ consists of merging two children nodes one level below as

$$\mathbf{g}_j^i = \text{TreeCell}_{\boldsymbol{\theta}}^{row}(\mathbf{g}_{2j-1}^{i-1}, \mathbf{g}_{2j}^{i-1}). \tag{16}$$

Here, $0 \leq i \leq \lfloor \log(u - 1) \rfloor + 1$ and $1 \leq j \leq \lfloor \frac{u}{2^i} \rfloor$, where $\mathbf{g}_j^0$ denotes the bottom-up summary state of $\mathcal{T}_j$. To compute a representation of all previously generated rows at each iteration, the model iteratively applies another Tree-LSTM cell over the relevant summaries to produce the row-level summary state $\mathbf{h}_u^{\text{row}}$:

$$\mathbf{h}_u^{\text{row}} = \text{TreeCell}_{\boldsymbol{\theta}}^{\text{summary}}\left(\left[\mathbf{g}_{\lfloor \frac{k}{2^i} \rfloor}^i \text{ where } k \text{ \& } 2^i = 2^i\right]\right) \tag{17}$$

where & is the bit-level 'and' operator. Thus, the Fenwick topology tree enables BiGG to generate rows in an autoregressive manner: the hidden state $\mathbf{h}_u^{\text{row}}$, defined in equation equation 17, contains a history of all rows generated in $\mathbf{A}$ so far and is used as the initial state for constructing the next decision tree, $\mathcal{T}_{u+1}$. Similarly, during training, the binary structure of the Fenwick row tree allows the initialization of row states for each row of $\mathbf{A}$ to be computed in $\mathcal{O}(\log n)$ time.

### A.2.4 BiGG Training Procedure

BiGG divides the training procedure into four main components, which all run on $\mathcal{O}(\log n)$ time:

1. First, because all decision trees $\mathcal{T}_u$ are known a priori during training, bottom-up summaries are computed from the leaves to the root of each tree. Importantly, the summary state for each row is independent of those for other rows, allowing these computations to be performed in parallel.

2. Next, using the root-level summaries from each $\mathcal{T}_u$, the model computes internal nodes $\mathbf{g}_j^i$ of the Fenwick topology tree in a level-wise manner using equation 16.

3. Once the Fenwick topology tree is constructed, BiGG concurrently computes the row summary states $\mathbf{h}_u^{row}$ in parallel using equation 17.

4. Finally, given each row-level summary $\mathbf{h}_u^{\text{row}}$, the model computes all left and right edge existence probabilities in parallel, traversing from the root to the leaves of each tree, as defined in equation 15.

Finally, we note that for graph generation, the trees $\mathcal{T}_u$ must be constructed sequentially. However, each $\mathcal{T}_u$ can still be built in $\mathcal{O}(\log n)$ time, reducing the overall runtime from $\mathcal{O}(n^2)$ to $\mathcal{O}((n + m) \log n)$.

### A.3 Adjacency-LSTM Motivation

The primary issue with using an LSTM to build the adjacency matrix of a graph is that most recurrent neural networks are best suited for linear data. Flattening an adjacency matrix and using an LSTM is one potential avenue for building a model, but suffers from significant drawbacks – the flattened vector varies based upon the node ordering $\pi$, and the model sacrifices the underlying structure of $\mathbf{A}$. (You et al., 2018)

---

**Algorithm 2** Adjacency-LSTM Sampling Algorithm

---

**Input:** Number of nodes $n$
1: **Initialization** Initial row node state $\mathbf{s}_{0,0}^R = (\mathbf{h}_{0,0}, \mathbf{c}_{0,0})$
2: **for** $i = 1, ..., n$ **do**
3:   $s_{i0} = s_{i-1,i-1} + \text{Pos}(n + 1 - i)$ {initialize new row node state}
4:   **for** $j < i$ **do**
5:    $\mathbf{s}_{ij} = \text{Cat}(\mathbf{s}_{i,j-1}^R, \mathbf{s}_{i-1,j}^C)$ {concatenate previous node states of row $i$ and column $j$}
6:    $p_{ij} = \sigma(f_p(\mathbf{h}_{ij}))$
7:    Sample edge $e_{ij} \sim \text{Bernoulli}(p_{ij})$
8:    **if** edge exists **then**
9:     $\mu_{ij} = f_\mu(\mathbf{h}_{ij})$
10:     $\log \sigma_{ij}^2 = f_{\sigma^2}(\mathbf{h}_{ij})$
11:     $\epsilon_{ij} \sim \text{Normal}(\mu_{ij}, \sigma_{ij})$
12:     $w_{ij} = \log(1 + \exp(\epsilon_{ij}))$
13:    **else**
14:     $w_{ij} = 0$
15:    **end if**
16:    $\text{embed}(e_{ij}, w_{ij}) = \text{Cat}(E_{ij}, f_w(w_{ij}), \text{Pos}(n + 1 - i), \text{Pos}(n - 1 + j))$
17:    $\mathbf{s}_{ij}^* = \text{LSTM}(\text{embed}(e_{ij}, w_{ij}); \mathbf{s}_{ij})$ {update adjacency state with edge embedding}
18:    $\mathbf{s}_{ij}^R, \mathbf{s}_{ij}^C = \text{Split}(\mathbf{s}_{ij}^*)$
19:   **end for**
20:   $\mathbf{s}_{i,i}^R = \mathbf{s}_{i,i-1}^R$ {set final row node state for subsequent row generation}
21: **end for**
**Output:** $G$ with $V = \{1, 2, ..., n\}$ and $E = \{e_{ij}, w_{ij}\}_{i=1; j>i}^n$

---

Generative models such as GraphRNN and GRAN, which use recurrent neural networks to build generative models of graphs, circumvent this issue by using node-level and graph-level recurrent networks that maintain edge generation and the global structure of the graph, respectively. Adj-LSTM was inspired by such methods and instead uses a partitioning of the hidden state to take advantage of the grid structure of the adjacency matrix directly. We will also show that partitioning the hidden state of a single LSTM provides greater generative quality, as this facilitates information passing between the states of the row and column nodes.

To adapt the LSTM architecture to a two-dimensional adjacency matrix, we partition the hidden state of the LSTM into $\mathbf{h}_{ij} = \begin{bmatrix} \mathbf{h}_{i,j-1}^R \\ \mathbf{h}_{i-1,j}^C \end{bmatrix}$, where $\mathbf{h}_{i,j-1}^R$ and $\mathbf{h}_{i-1,j}^C$ are the prior hidden states of the row and column nodes corresponding to entry $A_{ij}$. As all state updates are the same regardless of which entry $A_{ij}$ is being generated, we drop the subscripts $i$ and $j$ moving forward.

We re-compute the linear recurrence (4) using this partitioning of the hidden state. First, we note the dimensions of each weight and bias vector. Suppose that the hidden dimension is $h_{dim}$ and the embedding dimension is $i_{dim}$. Then we have the following:

1. $\mathbf{h}^R, \mathbf{h}^C \in \mathbb{R}^{h_{dim}} \implies \mathbf{h} \in \mathbb{R}^{2h_{dim}}$ and $\mathbf{W}_h \in \mathbb{R}^{2h_{dim} \times 2h_{dim}}$.

2. $\mathbf{x}_{ij} \in \mathbb{R}^{i_{dim}} \implies \mathbf{W}_i \in \mathbb{R}^{2h_{dim} \times i_{dim}}$.

3. $\mathbf{b} \in \mathbb{R}^{2h_{dim}}$.

Hence, we can partition the weight matrices and bias vector from the LSTM equations by defining the following partitions for each weight matrix:

$$\mathbf{W}_x = \begin{bmatrix} \mathbf{U}^R \\ \mathbf{U}^C \end{bmatrix} \tag{18}$$

Table 4: Performance on updating the states simultaneously ("Joint") vs separately ("Independent")

| Update Mode | Deg. | Clus. | Top Spec. | Wt. Spec. | MMDWt | Error |
|:---:|:---:|:---:|:---:|:---:|:---:|:---:|
| Joint | $2.46e^{-4}$ | 0.0 | $9.98e^{-4}$ | $8.51e^{-4}$ | $4.93e^{-3}$ | 0.065 |
| Independent | $3.27e^{-4}$ | $6.20e^{-5}$ | $2.98e^{-3}$ | $2.46e^{-3}$ | $6.40e^{-3}$ | 0.275 |

$$\mathbf{W}_h = \begin{bmatrix} \mathbf{V}^{RR} & \mathbf{V}^{RC} \\ \mathbf{V}^{CR} & \mathbf{V}^{CC} \end{bmatrix} \tag{19}$$

where each $\mathbf{U}^* \in \mathbb{R}^{h_{dim} \times i_{dim}}$ and each $\mathbf{V}^{**} \in \mathbb{R}^{h_{dim} \times h_{dim}}$.

Using the partitioning from 18 and 19 and partitioning the bias vector as $\mathbf{b} = \begin{bmatrix} \mathbf{b}^R \\ \mathbf{b}^C \end{bmatrix}$, we can partition the LSTM update equations as

$$\begin{bmatrix} \hat{\mathbf{h}}^R \\ \hat{\mathbf{h}}^C \end{bmatrix} = \begin{bmatrix} \mathbf{U}^R \mathbf{x} + \mathbf{V}^{RR} \mathbf{h}^R + \mathbf{V}^{RC} \mathbf{h}^C + \mathbf{b}^R \\ \mathbf{U}^C x + \mathbf{V}^{CR} \mathbf{h}^R + \mathbf{V}^{CC} \mathbf{h}^C + \mathbf{b}^C \end{bmatrix}.$$

Jointly updating the row and column states allows for the transfer of information between the row and column nodes via the weight matrices $\mathbf{V}^{RC}$ and $\mathbf{V}^{CR}$, which we hypothesized would mitigate the issue of long-term memory – as the model is predicting entries row-wise, information early in the row-generation process becomes lossy without the joint update property of the LSTM with a partitioned hidden state.

To test this hypothesis, we trained the lobster graphs on two models: one which uses the single LSTM-update on the concatenated row and column states, and the other which uses two LSTMs that update the row and the column states independently, which corresponds to setting $\mathbf{V}^{CR} = \mathbf{V}^{RC} = \mathbf{0}$ in Equation 19. As observed in Table 4, the LSTM joint update provides superior results on all observed metrics.

### A.4 HW-Tree Weight Generation

The hierarchical sampling scheme of the HW-tree weights provided a means of testing for autoregressiveness in the models with respect to the edge weights. There are two main quantities of interest: the global variance of weights pooled across all HW-trees, $\text{Var}(w_{ij})$, and the variance of weights found in a single HW-tree, $\text{Var}(w_{ij}|\mu_k))$. Note a few preliminaries that are easily derived from their respective distributions

1. $\mu_k \sim \mathcal{U}(7, 13) \implies \mathbb{E}(\mu_k) = 10$ and $\text{Var}(\mu_k) = 3$

2. $w_{ij} \sim \Gamma(\mu_k^2, \mu_k^{-1}) \implies \mathbb{E}(w_{ij}|\mu_k) = \mu_k$ and $\text{Var}(w_{ij}|\mu_k) = 1$

Importantly, the variance of the weights found in each tree is free of the parameter $\mu_k$. Next, an application of iterative expectation and variance yield the mean and variance of weights pooled from all trees as

1. $\mathbb{E}(w_{ij}) = \mathbb{E}_\mu[\mathbb{E}_w(w_{ij}|\mu_k)] = \mathbb{E}_\mu(\mu_k) = 10$.

2. $\text{Var}(w_{ij}) = \mathbb{E}_\mu[\text{Var}_w(w_{ij}|\mu_k)] + \text{Var}_\mu[\mathbb{E}_w(w_{ij}|\mu_k)] = \mathbb{E}_\mu(\mathbb{1}) + \text{Var}_\mu(\mu_k) = 1 + 3 = 4$

Thus, to test for autoregressiveness in the models, we observe that weights pooled from all HW-trees have variance $\text{Var}(w_{ij}) = 4$, whereas weights from a single HW-tree have variance $\text{Var}(w_{ij}|\mu_k) = 1$.

### A.5 Further Training Details

**Hyperparameters**  For Adj-LSTM, node states were parameterized with a hidden dimension of 128 and use a 2-layer LSTM. An embedding dimension of 32 was used to embed edge existence, and an embedding dimension of 16 was used to embed the weights. Positional encoding was used on the initialized row states.

For all models using BiGG, we use a hidden dimension of 256 with position encoding on the row states, as used in the original BiGG model. The weight state we use for BiGG-E has a hidden dimension of 16 for model runs and 32 for scalability runs. For BiGG-E and BiGG+GCN, bits compression of 256 was used on the topology state.

**Training Procedure**  Both models were trained using the Adam Optimizer with weight decay and an initial learning rate of $1e^{-3}$. The learning rate is decreased to $1e^{-5}$ when training loss plateus. Separate plateus are used for the weight parameters and the topology parameters.

- For the lobster data set, we train BiGG-E and BiGG-MLP for 250 epochs and validate at the 100 and 200th epochs. We plateau weight at epoch 50 and topology at epoch 100. We train Adj-LSTM for 300 epochs and validate every 100 epochs. We decay the learning rate at the 25 and 100th epochs.
- For the HW-tree data set, we train BiGG-E and BiGG-MLP for 550 epochs and validate every 100 epoch. We plateu weight at epoch 150 and topology at epoch 200. We train Adj-LSTM for 100 epochs and validate every 25 epochs, where the learning rate is plateaued at epochs 25 and 50.
- For the Erdős–Rényi data set, BiGG-E was trained for 500 epochs and validated every 250 epochs. We plateu weight at epoch 100 and topology at epoch 500. Due to slow training and poor convergence, the Adjacency-LSTM was only trained for 27 epochs.
- For the 3D Point Cloud data set, BiGG-E was trained for 3000 epochs and validated every 1000 epochs. We plateu weight at epoch 500 and topology at epoch 1500. Adj-LSTM was reported out of memory for this dataset.
- For the PT-tree graphs, BiGG-E was trained for 1000 epochs. We plateu weight and topology at 500 epochs.

BiGG-MLP and BiGG+GCN follow the training protocol for BiGG-E. For the convolutional network, two convolutions were used and each component was trained jointly on the same objective function used on BiGG-E.

**Baseline Models**  The Erdős–Rényi model baseline estimates were generated by first estimating the global probability of an edge existing between two nodes based on the training data, and then constructing Erdős–Rényi graphs with that probability of edge existence, as done in (You et al., 2018). Weights were sampled with replacement from all possible training weights in order to produce weighted graphs.

**Regularization**  We use the following to regularize our models, noting there is a propensity for the models to overtrain on the edge weights and decrease generative quality of the graphs. We believe this is directly related to the issue of balancing two losses - the topological and weight losses - which is further compounded by the fact learning the topology of a graph is much more challenging than the edge weights due to super-exponentially growing configuration space of graphs with respect to the number of nodes $n$. We note while BiGG-E had a tendency to overtrain on edge weights when plateauing the loss, the effect was much more prominent with BiGG-MLP, and regularization attempts were less successful with BiGG-MLP than on BiGG-E. As such, we use the following regularization on all models:

- We use weight decay on the Adam optimizer with decay $1e^{-4}$ on topology parameters and $1e^{-3}$ on weight parameters.
- Prior to inputting the sampled weight through the embedding LSTM, we standardize the weight as $\hat{w} = s_w^{-1}(w - \bar{w})$, where $\bar{w}$ and $s_w$ and the mean and standard deviation of all training weights, respectively. We note this is because the model is general purpose and must handle weights of varying magnitudes, where larger weights can potentially saturate the output of the embedding LSTM.

- The loss for weight is scaled down by a factor of 10 to balance the topology and weight losses. Upon plateauing both sets of parameters, the scale was increased to a factor of 100 for all graphs except the PT-tree graphs.
- When both losses were plateaued, the weight loss was updated every other epoch instead of every epoch to allow more fine-tuning of the graph topology without encouraging overtraining on the edge weights.

### A.6  Further Experimental Setup Details

Here, we provide additional details on our experimental setup. First, we describe the metrics used in detail, followed by controls used to ensure fair comparison across models.

**Evaluation Metrics**   We divide our metrics into three main groups: topology-only metrics, weight-only metrics, and joint metrics. We describe each metric in more detail here. First, we consider our topology-only metrics, noting these metrics are identical to the ones used in Liao et al. (2019), You et al. (2018), and Dai et al. (2020):

- **MMD on Degree (Deg.)**: distributions of unweighted degrees are computed for each graph. We use the Gaussian Total Variation kernel when computing the MMD (Liao et al., 2019).
- **MMD on Cluster (Clus.)**: distributions of unweighted clustering coefficients are computed for each graph. We use the Gaussian Total Variation kernel when computing the MMD.
- **MMD On Orbit**: distributions of the number of occurrences of orbits with 4 nodes (Liao et al., 2019).
- **MMD on Spectrum (Spec.)**: distributions of the spectral (set of eigenvalues) of each graph's normalized unweighted Laplacian matrix are computed. We use the Gaussian Total Variation kernel when computing the MMD.
- **Error**: this represents the percentage of graphs that do not hold a global graph property. For example, in the tree and yeast dataset, this is the percentage of graphs that are not true bifurcating trees with the correct number of leaves.

Next, we consider our weight-only metrics:

- **Global mean ($\bar{w}$)**: We pool all edge weights from all graphs to compute the global mean weight generated by each model.
- **Global standard deviation ($s_w$)**: We pool all edge weights from all graphs to compute the global standard deviation generated by each model.
- **Within tree standard deviation ($s_T$)**: For the tree dataset, we compute the standard deviation of weights found within each tree and then compute the mean of those standard deviations.
- **MMD on Marginal Weights (MMDWT)**: We consider the marginal distribution of weights per graph. We use the Gaussian earth mover distance (EMD) (Liao et al., 2019) when computing the MMD, as we found the EMD was a distance metric capable of considering outliers in the edge data sets. We found that the wildly different scales edge weights were on led to sensitivies to the MMDs, so the bandwidth $\sigma$ was tuned to an appropriate value using random partitions of the training graphs for each graph dataset, as needed.

Finally, we consider our joint metrics:

- **MMD on Weighted Spectrum (Spec.)**: distributions of the spectral (set of eigenvalues) of each graph's normalized *weighted* Laplacian matrix are computed. We use the Gaussian Total Variation kernel when computing the MMD.
- **MMD on Weighted Degree (Wtd. Deg.)**: distributions of unweighted degrees are computed for each graph. We use the Gaussian Total Variation kernel when computing the MMD.

Finally, we report the rank of each model within and across all datasets. The rank is computed by comparing the various metrics from the set {Mean - SD, Mean, Mean + SD}. For the global rank, ranks across datasets are pooled. Results are reported as Mean Rank $\pm$ SD Rank.

**Model Comparison**   To ensure fair comparison between all models when running experience, we implemented the following controls:

- We ran each model using the same 5 seeds and reported the mean statistic and standard deviation ans mean $\pm$ standard deviation.
- Each model was trained on and compared with the same set of training and test graphs, respectively.
- All MMD computations used the same hyperparameters and kernels.

