# OpenReview forum: "Scalable Generative Modeling of Weighted Graphs"
_TMLR — Accepted by TMLR_

### Review · Reviewer_cekc · 2025-08-06

**Summary Of Contributions:**

This work targeted on a scalable graph generation problem where graphs may contain weighted edges. Specifically, it proposes BIGG-E, built on top of a previous research work BIGG. To summarize, the contributions are as follow:
1. This work proposes a new graph generative model adopted from BIGG to tackle the weight graph generation problem.
2. Experiments support the claims that BiGG-E demonstrate better performance in generating large weight graphs.

**strengths**:
1. The paper is well structured and have clearly demonstrated the target problem to be solve
2. The proposed component that enable BiGG to model weight graph, BiGG-E, is novel and the design is reasonable.

**weakness**:
1. While the proposed component is novel, it is a modification on top of an existing model. I'd like to see a better justification of why it is designed in that way rather than other model architecture choices.
2. The overall contributions of the work are insufficient for a long paper submission, I suggest the authors considers a short paper venue or a workshop submission
3. The work propose datasets for modeling and evaluation, specifically, it turns the graph datasets used in BIGG into weight graph datasets. Moreover, the edge weight are assigned by sampling from some simple distribution (normal/uniform/etc...) and with some simple transform. While the experiment demonstrate that BiGG perform well on the modified datasets. It not clear this is a valid assessment. I will suggest the authors try their methods on some real weighted graph datasets.

**Audience:**

Yes

**Audience Explanation:**

Weight graph is common in real world but to my knowledge there are limited research working on this. Such work could be interesting to some audience working on this area

**Claims And Evidence:**

No

**Claims Explanation:**

My major concern is on the evaluation -- again since to me the edge weight distribution may be too simple, it doesn't necessarily support the validity of the propose components.

**Requested Changes:**

Better weight graph datasets should be introduced to further prove the effectiveness of the propose method.

---

> ### Author Response · Authors · 2025-08-21
> **Response**
>
> Thank you, cekc, for taking the time to review our work. We are pleased to hear that you found that “the paper is well structured and have clearly demonstrated the target problem to be solved,” and that the BiGG-E design is novel and reasonable. Below, we will address your main comments and concerns.
>
> > While the proposed component is novel, it is a modification on top of an existing model. I'd like to see a better justification of why it is designed in that way rather than other model architecture choices.
>
> We have provided a detailed justification for our architectural choices for BiGG-E in the general rebuttal, under the section “Motivation and Contribution.” To summarize: we chose to extend BiGG-E in our paper in such a way that we preserve the scalability of its unweighted predecessor, BiGG [1], as shown empirically in Figure 5 (p. 14). In particular, **the construction of the weight state using the Fenwick tree structure is what allows BiGG-E to train and generate weighted graphs with the same time complexity as BiGG on unweighted graphs.** If anything remains unclear, we are happy to address and follow-up comments.
>
> > The work propose datasets for modeling and evaluation, specifically, it turns the graph datasets used in BIGG into weight graph datasets. Moreover, the edge weight are assigned by sampling from some simple distribution (normal/uniform/etc...) and with some simple transform. While the experiment demonstrate that BiGG perform well on the modified datasets. It not clear this is a valid assessment. I will suggest the authors try their methods on some real weighted graph datasets.
>
> We agree that real-world datasets are of interest. With this in mind, **we have included results from an additional dataset, real-world phylogenetic (yeast) trees.** As mentioned in the general rebuttal under “graph datasets”, not all of our datasets rely on i.i.d. weight assignments from simple distributions. In addition, the 3D point cloud graphs are real-world datasets: they are reconstructed from household objects using a k-nearest neighbor algorithm, with edge weights defined as Euclidean distances in 3D space [2].
>
> > Requested Changes: Better weight graph datasets should be introduced to further prove the effectiveness of the propose method.
>
> We have added results on real-world phylogenetic trees (yeast dataset).We again note that the 3D point cloud data arises from collections of real objects arranged in space.
>
>
> > The overall contributions of the work are insufficient for a long paper submission, I suggest the authors considers a short paper venue or a workshop submission
>
> Thank you for your suggestion. We are receptive to submitting a moderately shorter version of the paper, should that avenue be a possibility in resubmission.
>
> [1] Hanjun Dai et al. Scalable deep generative modeling for sparse graphs. In Proceedings of the 37th International Conference on Machine Learning (ICML), volume 119 of Proceedings of Machine Learning Research, pp. 2302–2312. PMLR, 2020.
>
> [2]  M. Nermann et al. Graph Kernels for Object Category Prediction in Task-Dependent Robot Grasping. In Proceedings of the Eleventh Workshop on Mining and Learning with Graphs (MLG–2013), Chicago, US, 2013.

---

### Review · Reviewer_UjLQ · 2025-08-07

**Summary Of Contributions:**

This manuscript proposes a graph generative model that builds upon the autoregressive BiGG model, with a focus on weighted graph data. Benchmark experiments show some improvements over the chosen baselines. However, I have major concerns regarding the novelty of the method and the comprehensiveness of the performance evaluation, which affect the overall scientific contribution of this manuscript.

**Audience:**

No

**Audience Explanation:**

While the manuscript presents an attempt to extend the BiGG model to weighted graph generation, it currently appears underdeveloped in several key areas. The level of methodological innovation is limited, the writing and structure need refinement for better clarity, and the experimental evaluation lacks depth and comprehensive baseline comparisons. Given these concerns, I feel that the manuscript, in its current form, does not yet meet the standards expected for publication at TMLR as an original research contribution. I encourage the authors to further strengthen the novelty, improve the clarity of presentation, and conduct more thorough experiments to better support their claims.

**Claims And Evidence:**

No

**Claims Explanation:**

## 1. Insufficient baseline comparisons

There are many recent and powerful graph generative models published in the research community. Many of them have conducted experiments on molecular or vision weighted graph datasets—such as those listed in references [1–6]—which include discrete edge labels, and some have also experimented with data containing continuous edge labels, such as [2]. The authors mention and discuss some of these papers but do not compare their model’s performance against them. Even if the paper specifically focuses on weighted graph generation, it would still be worthwhile to compare against variants of these baselines to provide a fair and comprehensive assessment of model performance.

## 2. Lack of novelty in model design
The proposed BiGG-E model closely resembles the original BiGG model in many aspects, including its reliance on the Fenwick data structure. It appears to be more of an engineering refinement of the original model, rather than one that introduces substantial methodological innovation.

## 3. Limited relevance of benchmark datasets
Most of the benchmark datasets used are standard plain graph datasets with stochastically generated (synthetic) edge labels. This setup does not effectively demonstrate the practical value of modeling weighted graphs, nor does it convincingly show that the model captures real-world data distributions.

Reference:

[1] SwinGNN: Rethinking Permutation Invariance in Diffusion Models for Graph Generation

[2] Joint generative modeling of scene graphs and images via diffusion models

[3] Digress: Discrete denoising diffusion for graph generation

[4] Score-based generative modeling of graphs via the system of stochastic differential equations

[5] Pard: Permutation-Invariant Autoregressive Diffusion for Graph Generation

[6] Equivariant Denoisers Cannot Copy Graphs: Align Your Graph Diffusion Models

**Requested Changes:**

Please address concerns regarding the model's novelty and the performance evaluation protocol to strengthen the contributions.

The writing and structure of the paper can also be improved for better clarity and readability. Please rephrase the second setence of the abstract for clarity: "However, most current deep generative models are either designed for unweighted graphs and are not easily extended to weighted topologies or incorporate edge weights without consideration of a joint distribution with topology."

---

> ### Author Response · Authors · 2025-08-21
> **Response**
>
> Thank you, UjLQ, for taking the time to review our work and provide us with detailed feedback. Below, we will address your main comments and concerns.
>
> > Even if the paper specifically focuses on weighted graph generation, it would still be worthwhile to compare against variants of these baselines to provide a fair and comprehensive assessment of model performance.
>
> The architectures used in graph generative models are indeed diverse, and recent diffusion approaches have shown impressive results. However, a central contribution of our work is **scalability to very large graphs up to 15K nodes**, which significantly narrows the set of candidate comparisons. Among the references you provided, [2] and [6] fall outside the scope of this paper, and [1] and [3] have $O(n^2)$ complexity, which prevents scaling to large graphs. For comparison, we included SparseDiff [1] in our scalability ablation (Figure 5, p. 14) as a diffusion comparison model. Our results indicate SparseDiff becomes computationally infeasible beyond ~2K nodes. For this reason, we focused on autoregressive models such as BiGG. If you believe a broader set of comparisons is still necessary, we are open to further discussion.
>
> > The level of methodological innovation is limited [lack of novelty]
>
> We respectfully disagree with this claim. As discussed in the general rebuttal under "Motivation and Contribution", the architectural considerations of BiGG-E were made with the explicit goal of maintaining the training and sampling speed-ups observed in BiGG [1]. Furthermore, to our knowledge, there is limited work on scalable weighted graph generation. In particular, there are no general autoregressive graph generative models that accomplish such a feat. While our contribution is targeted, we maintain it is methodologically novel. We also note that other reviewers recognized this: Reviewer cekc described BiGG-E’s weight-generation component as “novel” and “reasonably designed”, and Reviewer EaE3 noted that the paper is “clear and properly motivated.” **Thus, we kindly ask you to reconsider the novelty aspect of our contribution. If you still disagree, we would appreciate clarification on what you feel is missing for our paper to be considered fully novel.**
>
> > Most of the benchmark datasets used are standard plain graph datasets with stochastically generated (synthetic) edge labels.
>
> As noted in the general rebuttal under "Graph Datasets", we have added real-world phylogenetic (yeast) trees to our experiment and clarified that 3D point clouds are another example of real-world data.
>
> > and the experimental evaluation lacks depth and comprehensive baseline comparisons
>
> We kindly ask you to clarify how our experimental evaluation lacks depth. The purpose of each model is to ensure a comparison between the various components of BiGG-E (See 3.5 Comparison Models, p. 9). For example, ADJ-LSTM is a slower comparison used to demonstrate the scalability gains of BiGG-E, BiGG-MLP is a simple extension that is unable to model both topology and edge weights, and BiGG+GCN is a two-stage model used to demonstrate BiGG-E's superiority in jointly modeling edge weights.
>
> Furthermore, our evaluation protocol uses a variety of statistics aimed at fully summarizing the quality of weighted graphs (see 4. Experiments, p. 9). The topological metrics included MMD on various attributes of the graphs that are standard metrics used when evaluating graph generative models ([1-3]). Furthermore, we provide more metrics to also evaluate the marginal distribution of edge weights, as well as the joint distribution of edge weights and topology.
>
> > Please address concerns regarding the model's novelty and the performance evaluation protocol to strengthen the contributions.
>
> We have now clarified our contribution and why we believe the contribution is novel. We have also strengthened our performance evaluation by including more real-world examples in our experimental analysis.
>
> > The writing and structure of the paper can also be improved for better clarity and readability.
>
> We have revised the abstract sentence into two shorter sentences for clarity (p. 1 of revision). Regarding the broader comment on the writing and structure, we would again appreciate clarification on which specific parts of the manuscript were unclear, noting that the other reviewers found the paper to be clearly structured and well-motivated. Your guidance would be valuable in helping us strengthen the presentation with targeted improvements.
>
> [1] Yiming Qin et al. Sparse training of discrete diffusion models for graph generation, 2024.
>
> [2] Hanjun Dai et al. In Proceedings of the 37th International Conference on Machine Learning (ICML), volume 119 of Proceedings of Machine Learning Research.
>
> [3] Renjie Liao et al.  Efficient graph generation with graph recurrent attention networks. In Hanna M. Wallach, Hugo Larochelle, Alina Beygelzimer, Florence d’Alché Buc, Emily B. Fox, and Roman Garnett (eds.), NeurIPS.

---

> ### Comment · Reviewer_UjLQ · 2025-08-26
>
> Thank you for your rebuttal, and I appreciate your efforts in providing additional results and drafting detailed responses. I also thank you for including real-world datasets, which partially address my earlier concern.
>
> ## On Experimental Validation Rigor
>
> In Table 1, both in the original and current versions, relatively small-scale datasets such as Tree and Lobster are used. As you note, these datasets contain no more than 200 nodes in their largest graphs. If extremely large graphs cause out-of-memory issues on your computing setup, these smaller sizes should still be manageable for running more recent baselines.
>
> In the current version, you include Adj-LSTM, two BiGG-based baselines (BiGG-MLP and BiGG+GCN), and the traditional stochastic method Erdős–Rényi for comparisons. However, in the reference work [1] (Yiming Qin et al.), the authors evaluate more than five recent deep generative methods in their main Tables 1 and 2, ensuring a comprehensive comparison against the latest approaches. This field is advancing rapidly, and to make the scientific contribution more convincing, it is important to demonstrate clear performance gains against more recent deep generative models on graphs.
>
> ## On Methodology
>
> I would also like to emphasize, in agreement with Reviewer cekc, that the methodological novelty remains a concern. The proposed method appears to be a modification of an existing approach and reuses the Fenwick data structure in the algorithm. While I fully acknowledge that the problem of generating large-scale graphs is meaningful and provides strong motivation, this alone does not sufficiently establish the novelty of the proposed method.
>
> Therefore, I retain the same overall assessment as in my initial review.

---

> > ### Author Response · Authors · 2025-08-26
> >
> > Thank you for your follow-up, and we are glad that the inclusion of real-world datasets was well received.
> >
> > We would like to re-iterate that our primary contribution is to provide **a model that learns a joint distribution over weighted graphs while scaling to large graph data.** We agree that additional comparisons to recent baselines on small-scale datasets could further strengthen the study and appreciate this suggestion for future work. At the same time, according to TMLR’s acceptance guidelines (https://jmlr.org/tmlr/acceptance-criteria.html), a submission should not be rejected simply because it does not achieve a new state-of-the-art on benchmark datasets. While further comparisons would be valuable, they are not necessary to support our work’s contribution, especially given the absence of true competitors.
> >
> > Regarding novelty, to our knowledge, no prior work has directly studied this problem, and there are no existing autoregressive models for weighted graphs that consider a joint distribution with topology and edge weights. While the contribution is targeted, it meaningfully extends graph generative modeling into weighted graph settings. We also note that Reviewer cekc explicitly described the method as novel while requesting justification of the design, which we addressed in the rebuttal, and Reviewer EaE3 called the paper “highly interesting to the graph generative community.” Both reviewers also responded “yes” when asked whether TMLR’s audience would find the findings of this paper of interest; regardless, TMLR explicitly states that “novelty of the studied method is not a necessary criteria for acceptance.”
> >
> > **In summary, we develop a method that meets a previously unmet need, and our experimental design directly evaluates the claims in our paper and provides a contribution that TMLR’s audience will find of interest, thus meeting the acceptance criteria outlined by the journal.**
> >
> > We thank you for your continued consideration of our manuscript.

---

### Review · Reviewer_EaE3 · 2025-08-12

**Summary Of Contributions:**

In this work, the authors extend the BiGG model for generating large-scale unweighted graphs to BiGG-E, generating weighted graphs while preserving the scalability, and in particular, the runtime complexity, of BiGG. The authors leverage the Fenwick tree construction from BiGG to also sample edge weights but do this in a joint modeling approach which allows to condition an edge weight on both the previously generated topological and edge weight features. The authors demonstrate empirically that BiGG-E's joint modeling approach outperforms naive extensions of BiGG as well as decoupled generation of topology and edge weights and scales to graphs up to 15K nodes.

I find the paper to be clear and properly motivated but I have doubts about the statistical significance of the results (see below for details).

**Additional Comments:**

n/a

**Audience:**

Yes

**Audience Explanation:**

The paper is highly interesting to the graph generative community. In particular, the question of how to properly generate weighted graphs, and the investigation  into different approaches to do so, as done in this work, are of some significance to the field.

**Claims And Evidence:**

No

**Claims Explanation:**

To my understanding, the main claims of the paper are:

**C1**: *BiGG-E is an extension of BiGG to support weighted graphs while maintaining the scalability of BiGG.*

BiGG-E maintains the scalability of BiGG by leveraging the Fenwick tree also for the weights, essentially only adding one parallel computational stream to BiGG. The authors further demonstrate this empirically, showing that generation time, memory requirements, and training time between BiGG and BiGG-E are fairly similar.

**C2**: *BiGG-E's joint modeling approach is more effective than naive extensions of BiGG.*

The authors provide two more extensions of BiGG as baselines which I would agree are straightforward ways to incorporate edge weights into BiGG: Generating edge weights directly based on the current topological state (BiGG-MLP) and first generating the graph and then predicting the edge weights after (BiGG-GCN). BiGG-E appears to surpass both approaches but I have some doubts about the standard error of the numbers in Table 1 and 2: In Table 1, no error bars are given. Considering the small size of the training datasets and the small MMD scores of the BiGG baselines, I wonder whether these results are statistically significant. In particular, the authors should have repeated these experiments multiple times and report standard error of these scores. In Table 2, the authors appear to provide standard deviation (SD). However, the SD is so high that the results do not appear to be statistically significant. However, it could be that I am misunderstanding these scores. Unfortunately, the paper does not provide a lot of explanation to the exact computation and meaning of these scores.

**Requested Changes:**

Regarding my concerns on the validity of **C2** above, I recommend the following changes:

* The authors should provide more explanation to the computation and interpretation of the different MMD metrics computed for topological and edge weight quality of the compared models. In particular, the authors should make clear whether the SD in Table 2 refers to the standard deviation of the edge weight accuracy.

* The authors should repeat their experiments multiple times and provide error bars for their reported scores to asses whether the results are statistically significant.

---

> ### Author Response · Authors · 2025-08-21
> **Response**
>
> Thank you, EaE3, for taking the time to review our work. We are pleased to hear you found the paper to be clear and properly motivated, and that “the paper is highly interesting to the graph generative community.” Below, we will address your main comments and concerns.
>
> > I have some doubts about the standard error of the numbers in Table 1 and 2: In Table 1, no error bars are given. Considering the small size of the training datasets and the small MMD scores of the BiGG baselines, I wonder whether these results are statistically significant. In particular, the authors should have repeated these experiments multiple times and report standard error of these scores.
>
> Thank you for voicing your concern regarding the statistical robustness of our results. Regarding interpretation, the absolute magnitude of an MMD score is less meaningful, since it depends on kernel choice and bandwidth. Instead, the MMD statistics used here provide multiple **relative points of comparison across models under identical conditions.** To ensure fairness in our model comparisons, we applied the following controls: (1) each model generates the same number of graphs as the size of the test set; (2) all models are trained and evaluated on the same datasets; (3) for each run, we use identical seeds when sampling graphs to ensure that node-counts are identical prior to generating a graph; and (4) MMD hyperparameter choices (kernel, bandwidth) are fixed across all models.
>
> That said, we agree that MMD estimates are subject to statistical noise, which makes it challenging to tease out performance when models are performing similarly within some datasets. **To directly address this concern, we have repeated all BiGG-E experiments with 5 seeds and now report mean ± standard deviation across seeds in Tables 1 and 2 (p. 11-12).** We also report model rankings within and across datasets to more easily compare performance. These results confirm that BiGG-E still consistently outperforms or competes with the comparison models.
>
> > In Table 2, the authors appear to provide standard deviation (SD). However, the SD is so high that the results do not appear to be statistically significant. However, it could be that I am misunderstanding these scores. Unfortunately, the paper does not provide a lot of explanation to the exact computation and meaning of these scores.
>
> We understand the confusion that the reported SD measures may have caused.  In Table 2, the reported means and standard deviations are global first- and second-order statistics of the marginal edge-weight distributions across all graphs. For the hierarchical-weight tree dataset, we report a third quantity $s_T$, which is the average within-tree standard deviation of the edge weights. We agree this distinction was unclear in the manuscript and have made the following updates: (1) in Table 2, we now use consistent notation for the marginal means and standard deviations of the underlying distributions ($\mu_w$, \sigma_w) as well as those computed from the generated graphs ($\bar{w}$, $s_w$); and (2) we explicitly define these terms in the main text (p. 9). If further changes are warranted to make our results clearer, we would value your additional feedback.
>
>
> **Requested Changes**
>
> > The authors should provide more explanation to the computation and interpretation of the different MMD metrics computed for topological and edge weight quality of the compared models. In particular, the authors should make clear whether the SD in Table 2 refers to the standard deviation of the edge weight accuracy.
>
> We have updated Table 2 to improve clarity in the results as mentioned in your comment. We have also included a longer explanation of the MMD and other metrics used in the Appendix (A.6, p. 26-27). We hope these changes address your concern, and we would be glad to make further adjustments if additional clarification is needed.
>
> > The authors should repeat their experiments multiple times and provide error bars for their reported scores to assess whether the results are statistically significant.
>
> We agree with your suggestion and have revised the manuscript accordingly. In Tables 1, 2, and 3, we now report the mean ± standard deviation of each metric across 5 independent seeds for BiGG-E and all comparison models.  Please let us know if your concern has been addressed with our changes.

---

### Author Response · Authors · 2025-08-21
**General Rebuttal and Follow-ups**

We thank all reviewers for taking the time to review and provide feedback on our manuscript – we greatly appreciate the insights that have been provided. In this rebuttal, we first address the overarching themes raised across reviews and describe the corresponding revisions we have made. We then follow up and provide individual responses to each reviewer’s specific comments.

We are encouraged by the positive assessments that “the paper is well structured and has clearly demonstrated the target problem to solve” (cekc), that our work is “highly interesting to the graph generative community” (EaE3), and that  “the proposed component to enable BiGG to model weighted graphs, BiGG-E, is novel” (ceks).

**Motivation and Contribution (UjLQ, cekc)**

There were some questions and concerns regarding the motivation and contributions outlined in our work. Here, we hope to clarify these concerns.  BiGG-E was designed to simultaneously address two needs, with an emphasis on the second: (1) the lack of **generative models for weighted graphs**, and (2) the need for models that **scale to very large graphs** (thousands of nodes). Among existing approaches, BiGG [1] uniquely scales to graphs of such sizes, while most alternatives struggle with computational bottlenecks. Since BiGG is limited to unweighted graphs and only models topology, our extension fills this gap by enabling joint modeling of topology and edge weights, where BiGG-E empirically achieves superior performance over our comparison models. To our knowledge, **there is no work on scalable generative modeling of weighted graphs with an emphasis on joint modeling**, which makes our work a novel contribution.

For weight generation specifically, our priority was to preserve the scalability advantages of BiGG. Using other architectures such as graph neural networks or attention mechanisms would significantly increase training and sampling costs, undermining this goal. Instead, we adopted the Fenwick tree data structure to maintain the same time complexity as BiGG for both training and sampling (see Figure 5 p. 14). In contrast, comparison models that rely on other architectures (e.g., BiGG+GCN) failed to capture such dependencies effectively (see Table 2 p. 12, HW-Tree and PT-Tree Graphs). We believe this demonstrates the necessity and advantage of our chosen design.

**Graph Datasets (UjLQ, cekc)**

There were concerns raised about the lack of real-world example datasets in our experiments. We agree that providing results on more real-world examples would strengthen the results of our studies. Thus, **we have included results on real-world phylogenetic trees** (Yeast data [2]) to address this concern. Results are located in Tables 1 and 2 of the revised manuscript.

We would also like to clarify the datasets in our experiments. The 3D point cloud graphs are real-world, where graphs are reconstructions of various household objects, equipped with edge weights that are Euclidean distances [3]. The remaining synthetic datasets capture dependencies in weighted graphs. For example, the hierarchical-weight trees dataset (p. 13) uses a sampling scheme in weight generation to produce correlated edge weights, and our path-threshold trees construction (p. 13) provides an explicit example where topology depends on the edge weights.

**Statistical Validity of Results (EaE3)**

A reviewer mentioned that there were issues with our metrics, particularly with discerning between statistical noise and truly superior performance of the model. To address this concern, **we re-ran BiGG-E and all comparison models on all datasets using the same 5 seeds and now report all results as mean $\pm$ SD** (Tables 1 and 2, p. 11-12). We also include a ranking and improve the readability of our tables of results. We expand our explanation of the Maximum Mean Discrepancy (MMD) statistics used in our analysis, and provide clarification on the first and second order summary statistics of the marginal distribution of edge weights (p. 9). Finally, we provide more detail of our MMD and experimental set-up in the appendix (Appendix A.6, p. 26-27). Kindly note in the interest of time, error bars are currently not available for ADJ-LSTM on the larger graphs, but can be included in a revised manuscript.


[1] Hanjun Dai et al. Scalable deep generative modeling for sparse graphs. In Proceedings of the 37th International Conference on Machine Learning (ICML), volume 119 of Proceedings of Machine Learning Research, pp. 2302–2312. PMLR, 2020.

[2] Gabriel W. Hassler et al. Tolkoff, Andrew J. Holbrook, Guy Baele, Philippe Lemey, and Marc A. Suchard. Principled, practical, flexible, fast: A new approach to phylogenetic factor analysis. Methods in Ecology and Evolution, 13(10):2181–2197, 2022.

[3]  M. Neumann et al. Graph Kernels for Object Category Prediction in Task-Dependent Robot Grasping. In Proceedings of the Eleventh Workshop on Mining and Learning with Graphs (MLG–2013), Chicago, US, 2013.

---

### Decision · Action_Editor_xQav · 2025-09-27

**Recommendation:** Accept as is

**Additional Comments:**

N/A (see above)

**Audience:**

Yes

**Audience Explanation:**

This paper tackles the problem of generative graph modeling with weighted edges. It claims that jointly modeling edge weights together with the graph topology is advantageous. The paper can be of interest for researchers working on generative models of graphs.

**Claims And Evidence:**

Yes

**Claims Explanation:**

The main contribution of this paper is to develop an extension of BiGG, a autoregressive graph generative model, allowing it to model weighted graphs. The resulting model, called BiGG-E, provides a method to generate both the graph and the edge weights. Importantly, the paper emphasizes the approach retains the computational efficiency of the base BiGG model.

There are two main claims of this paper:

1. The proosed BiGG-E model supports weighted graphs as part of the generative process, while maintaining the scalability of BiGG. This is indeed the case, since BiGG-E also uses the Fenwick tree for the weights. In fact, the paper demonstrates empirically that the generation time, memory requirements, and training time of both BiGG and BiGG-E are similar.

2. Modeling the weights jointly with the graph's topology (like BiGG-E does) is more effective than naive extensions of BiGG. This is also supported by experiments.

During the review process, reviewers found that claims in the paper were not properly supported. Specifically:
- Baselines were insufficient (some models not considered).
- The model design is incremental, and not novel enough.
- The considered datasets were not enough to properly demonstrate BiGG-E's ability.
- Experiments did not include error bars.
In their rebuttal, the authors tried to address most of these concerns. In particular, certain baselines were incorporated (with the caveat that most of the competing models cannot scale as much), a new dataset was added, and error bars were included. Although the paper could perhaps be strenghtened in terms of the baselines considered, I believe these concerns have been properly addressed.

The main outstanding concern (shared by reviewers cekc and UjLQ) is about the novelty of the paper. The authors discuss in their rebuttal that novelty is not a criterion for TMLR acceptance. Thus, despite the work being considered somewhat incremental, I agree with the authors that this concern should be downweighted, and thus the paper is above the TMLR acceptance bar.